# Hydrodynamic Modeling and Comprehensive Assessment of Pier Scour Depth and Rate Induced by Wood Debris Accumulation

**Muhanad Al-Jubouri** * and **Richard P. Ray**

Department of Structural and Geotechnical Engineering, Faculty of Civil Engineering, Széchenyi István University, Egyetem tér, 9026 Győr, Hungary; ray@sze.hu
* Correspondence: al.jubouri.muhanad@hallgato.sze.hu or muhanad.kh.99.oo@gmail.com

**Abstract:** This study mainly investigates the impact of debris accumulation on scour depth and scour hole characteristics around bridge piers. Through controlled experiments with uniform sand bed material, the influence of various debris shapes (high wedge, low wedge, triangle yield, rectangular, triangle bow, and half-cylinder), upstream debris length, downstream debris extension, and debris thickness on scour depth and scour hole area and volume around the cylindrical pier were analyzed. The findings revealed that the shape and location of debris in the water column upstream of piers are key factors that determine the depth of scour, with high wedge shapes inducing the deepest scour and potentially the largest scour hole, particularly when positioned close to the pier and fully submerged. Scenarios in which triangle bow debris was submerged at full depth upstream of the pier closely resembled situations devoid of debris. Conversely, debris extension downstream of the pier was found to reduce local scour depth while concurrently enlarging the dimensions of the scour hole. The existing scour prediction equations tend to overestimate scour depth in scenarios involving debris, particularly when applying effective and equivalent pier width. This discrepancy arises because these equations were originally developed to predict scour depth around piers in the absence of debris. In response, a refined model for predicting scour induced by debris was proposed, integrating factors such as upstream debris length, downstream extension, obstruction percentage, and debris shape factor. This model demonstrated strong agreement with experimental data within the scope of this study and underwent further validation using additional experimental datasets from other research endeavors. In conclusion, this experimental study advances the comprehension of scour processes around cylindrical bridge piers, providing valuable insights into the role of debris characteristics and positioning.

**Keywords:** debris; pier scour; downstream extension length; deposition height; effective pier width





## 1. Introduction

Bridges are subject to various detrimental factors, such as water flow, sedimentation, and debris accumulation. Researchers have used the term large floating wood debris (*LFWD*) to describe log jams, debris rafts, drift, and debris masses, which typically consist of woody debris accumulating near bridge piers and pose a significant threat to bridge safety. The debris can increase lateral forces on the piers and reduce the flow area under the bridge, leading to higher flow rates and extensive scouring around the piers, thereby reducing the bridge's stability and contributing to possible collapse [1].

Over the past few decades, hydraulic engineering has seen significant advancements through computational and experimental investigations. These efforts have greatly improved the understanding of the scouring process around bridge piers, illuminating the underlying fluid dynamics and the formation of turbulent horseshoe vortices (*THV*) upstream of the bridge piers [2–8]. Kirkil et al. [9] reported that the horseshoe vortex (*HV*) system around a circular pier in a scour hole changes in location and time. There is also

a high average bed shear stress under the main horseshoe vortex and at the cylinder pier base. Misuriya et al. [10] demonstrated calculating the scour depth in low roughness ranges $(D/D_{50}) < 25$ with an error of less than 15% while 30% for roughness ratios $(D/D_{50} \geq 100)$ The study suggested that the flow/pier diameter ratio in gravel beds influences the scour depth, potentially assisting in bridge pier design. Studies such as [11–16] show that many researchers have investigated the temporal dynamics of peak scour development near bridge piers. Using field data and experimental insights, these researchers have developed several equations to predict scouring.

Flow conditions, stream geometry, pier configuration, and debris characteristics influence the shape and quantity of debris accumulation [17]. Debris can restrict the channel's cross-section, increasing flow velocities that exacerbate scouring and sediment removal [18]. Debris accumulation can vary in size and shape, ranging from small debris clusters upstream of a pier to near-complete blockages of the bridge's entrance channel [19]. Past studies [18] have identified two primary shapes for debris accumulations: rectangular and triangular. These shapes depend on elements such as the characteristics of transported debris and the waterway's shape. Cylindrical bridge piers, especially those wider width are particularly prone to debris clogging [20]. Pagliara and Carnacina [21] studied the effect of the shape and obstruction area of debris on the degree of pier scouring. They found that the obstruction area had a more significant effect than the frontal shape on the maximum scour hole depth. However, they only focused on three debris shapes, namely rectangular, triangular, and cylindrical, with a fixed upstream length that did not extend downstream of the pier. They later investigated the influence of the length of debris upstream and its extension downstream on pier scouring [22]. Their findings suggest that the maximum scour depth correlates with the debris length ratio to pier width, with the maximum scour depth occurring at a ratio of 3. Additionally, they observed that the scour depth decreases as the debris extends downstream from the pier.

Previous research has predominantly concentrated on a narrow range of debris shapes, such as rectangles and triangles, typically accumulated upstream of bridge piers. Moreover, existing studies have commonly assumed that debris collects on the free surface, mid-depth, or channel bed. However, researchers have paid limited consideration to the potential impact of the extension of floating debris resulting from continuous accumulation processes both upstream and downstream of the pier and downward toward the stream bed. The current experimental investigation systematically addressed the abovementioned constraints by examining the consequences of diverse debris shapes and obstruction levels on the cylindrical pier. The debris shapes included: a rectangle block, high wedge, triangle bow, low wedge, half-cylinder, and triangle yield sign. The experiments occurred under clear-water conditions, meticulously ensuring that the approach flow velocity remained below the threshold required to initiate particle motion. The experimental scenarios considered three positions of debris: floating in the upper 25% of the water's depth, extending to 50%, and extending to 100% of the flow depth.

Furthermore, each case systematically examined the downstream extension of debris, ranging from 0% to 33% to 66% of the upstream debris length. Specifically, the research examined scenarios where the debris obstruction percentage ($A\%$) covered up to 33% of the flow area and explored the impact of debris accumulation extending downstream up to 66% of the upstream length. This investigation used a narrow flume with a width of 0.3 m to maintain controlled conditions and precise monitoring.

Based on these findings, an enhanced predictive model was developed to estimate debris-induced scour. The model incorporates variables including upstream debris length, downstream extension, obstruction percentage, and debris shape factor. Through careful validation against experimental data from this study and supplementary datasets from Melville and Dongol [23], the model demonstrated exemplary performance and reliability within the datasets examined.

## 2. Experimental Characteristics and Sand Properties

All experimental procedures were conducted at the Research and Design Center Laboratory, Ministry of Water Resources, Iraq. The laboratory houses a dedicated testing environment comprising a horizontal flume of 12.5 m in length, 0.3 m in width, and 0.55 m in depth (Figure 1). Its transparent glass walls allow for direct observation and analysis of all 100+ experiments. An electric pump drove the hydraulic system to a maximum discharge rate of $Q$ = 90 L per second, which supplied the requisite flow to the experimental channel. The experimental setup incorporated specific components to ensure optimal flow characteristics and precise data collection. Upstream of the channel, a sluice gate provided flow straighteners featuring a mesh size of 0.2 cm. Downstream, a screen regulated and stabilized the flow conditions. Including a flow straightener at the flume's intake helped provide a consistent flow regime while mitigating surface wave disturbances, vortices, and turbulence that could arise from fluctuations in the pump operation. This system allowed for the recirculation of discharged water from the flume back to a sump and contributed to the overall efficiency of the experimental setup.

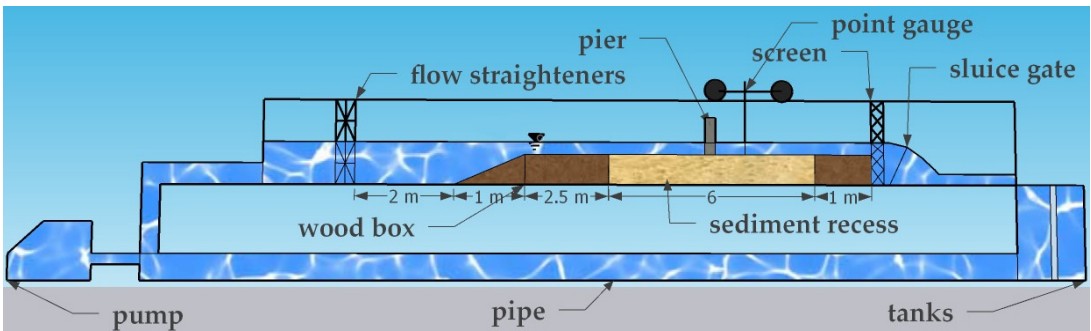

**Figure 1.** A schematic of the experimental flume setup illustrates the essential components used in the study.

The area devoted to scouring processes spanned a length of 6 m, a depth of 0.15 m, and a width of 0.3 m. An upstream wooden box measuring 3.5 m in length seamlessly transitioned from the inflow sluice to the scour zone. The depth was reduced from 0.55 m to 0.15 m using the 1.5 m-long inclined surface of the box. A thin layer of sediment was uniformly distributed over this box to maintain the continuity of bed roughness. A volumetric flow meter determined flow in the channel with a specified accuracy of $\pm1\%$. Furthermore, an electronic flow rate gauge installed in the flume discharge system offered precise measurements with an accuracy of $\pm0.05\%$. Within the sediment recess, where the principal investigations took place, the presence of the bridge pier and the underlying base material was a defining characteristic of the experimental scenario. The dimensional assessment and resemblance theory concepts served as a systematic framework for choosing the parameters of the experimental model.

A variety of dimensionless characteristics were carefully taken into account, such as those related to flow, flume, and sediment conditions. These characteristics describe the system's behavior independent of units, allowing the model to be seamlessly scaled to various scenarios, from small lab flumes to vast rivers. Based on studies by Oliveto and Hager [24] and Raudkivi and Ettema [25], testing should be accomplished using sediment with a median particle size $D_{50} > 0.7$–$0.8$ mm to avoid creating ripples. Thus, these experiments used a uniform, medium river sand with a $D_{50}$ = 0.93 mm. The angle of repose was 32 degrees. Sediment density ($\rho_s$) was 2650 kg/m$^3$, while the water density ($\rho_w$) measured 1000 kg/m$^3$. The sand possessed a geometric standard deviation ($\sigma = \sqrt{\frac{D_{84}}{D_{16}}}$) of 1.28, where $D_{16}$ and $D_{84}$ were diameters, which were 16%, and 84% of the sample was finer. This specific sediment choice was deliberate, yielding a particle size standard deviation of less than 1.3, ensuring minimal armoring effects [26,27]. The condition of uniform bed

material may not accurately predict scour depths in nature, as sediment gradation is typical, leading to differences in bed material composition. Field predictions often underestimate scour depths compared to assumptions of homogeneous bed material [27].

The authors of [28,29] suggested that the ratio of pier width (*D*) to channel width (*B*) should ideally be less than 10% to minimize the impact of flume walls on the scour depth. In this study's experiments, a cylindrical PVC pier was employed with a constant diameter (*D*) of 2 cm, representing less than one-sixth of the flume width (*B* = 30 cm). Furthermore, Yang and Lim [30] contributed valuable insights by determining that the essential aspect ratio that is equal to the flume width (*B*) to the flow normal depth (*Y*), (*B*/*Y*), a key parameter for analytical solutions governing shear distribution in smooth rectangular channels, assumes a value of 2. This study fulfills this assumption with an aspect ratio equal to 2.5. Finally, Raikar and Dey [31] explored the connection between water depth and the development of armor in riverbeds. They found that when the *Y*/*D* ratio exceeded 3 for nearly uniform bed material, the effect was negligible. These previous findings were taken into account during the selection of experimental parameters for the current study.

All experiments within this investigation adhered to a clear water sediment transport condition, where the ratio of mean/critical flow velocity (*V*/*Vc*) remained below 1.0. The entrainment velocity during tests conducted without piers closely aligned with Shields' diagram [32], the equations proposed in [33,34] yielded around $Vc = 0.32$ m/s and $V = 0.22$ m/s, considering a flow depth (*Y*) of 0.12 m, and the Froud number was equal to 0.203, which confirmed the subcritical flow. The Reynolds number, $Re_D$, characterizes the turbulence generated by the pier rather than the waterway itself, which was equal to $5.8 \times 10^4$. The authors of [35] emphasized its significant influence on the power of the horseshoe vortex, a key factor contributing to scour formation. However, according to [6], $Re_D$ has a small impact on the relative vortex dimensions. Therefore, the pier scour is more affected by the high $Re_D > 10^4$.

## 3. Large Wood Floating Debris

The initial phase of experimental design required understanding the behavior of debris lodged against bridge piers and its contribution to local scour [36]. All experimental scenarios shared some common properties:

- All debris clumps were impervious. Previous research [19,37] indicated that the porosity of debris accumulation has a negligible influence on scour hole depth and shape. However, it does influence the dynamic pressure on bridge piers.
- The debris clumps were fixed in position. Stationary debris allowed for a consistent geometry when comparing scour holes caused by varying other parameters.
- The flow direction angle remained at 0° concerning the debris' upstream face. Maintaining a normal direction eliminated the effects of the attack angle.
- The debris shapes could be submerged to different depths.
- The debris shapes could be positioned so that they extended upstream and downstream of the pier.

Importantly, this approach maintained consistency and reproducibility in experimental setups, aligning with similar methodologies documented in [17,19,23]. By adopting such a systematic and controlled approach, this study contributes valuable insights to the engineering community, offering a reliable foundation for further research and practical applications.

Figure 2 shows the various debris shapes used in the experimental program perpendicular to the flow direction. The debris' width and length (downstream) measured 0.12 m and 0.06 m. Notably, the height of the debris varied across three distinct settings: 0.03 m, 0.06 m, and 0.12 m, facilitating a detailed investigation into their individual effects. The depicted debris forms include:

- Rectangular block (*RB*): A block-shaped configuration.
- Triangle bow (*TB*): Resembling the bow of a ship, oriented to face upstream.

- High wedge (*HW*): Featuring a large mass at the high end, with the upstream face directing the flow downward.
- Low wedge (*LW*): Characterized by a large mass at the low end, with the upstream face directing the flow upward.
- Triangle yield sign (*TY*): Wide at the top with an apex at the bottom, featuring a flat upstream face.
- Half-cylinder (*HC*): Presenting a rounded side facing upstream.

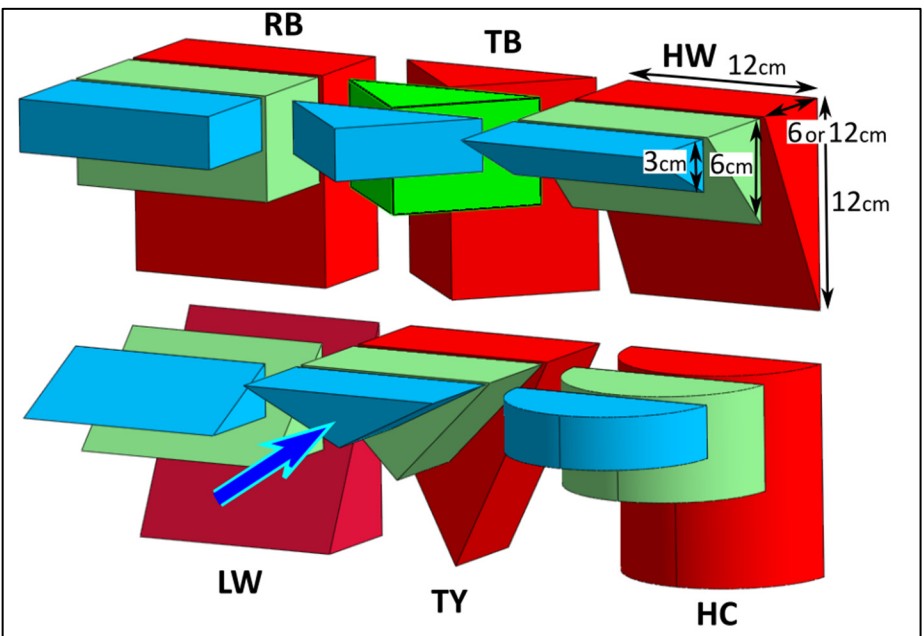

**Figure 2.** Varieties of debris shapes and dimensions, including rectangle block (*RB*), triangle bow (*TB*), high wedge (*HW*), low wedge (*LW*), triangle yield sign (*TY*), and half-cylinder (*HC*).

In this experiment, the selected forms closely mimic real-world debris accumulations around piers, considering both natural formations and experimental constraints. Notably, configurations such as rectangular blocks and low wedges resemble consolidated masses of debris with thin branch layers, reflecting observed patterns around piers. The shapes align with the findings of [19], highlighting the prominence of triangular-in-depth configurations in debris accumulation. The study focused on representative geometries, such as ideal inverted half cones, triangle bows, triangle yield signs, and high and low wedges. Additionally, the half-cylinder shape is relevant to prior works [23,38]. To evaluate debris accumulation dimensions, assessment values were derived from mean ranges in existing literature, primarily drawn from field surveys, i.e., [19], localized analyses, and established laboratory guidelines documented in [22,38,39], along with the test range developed in [23].

Key parameters under consideration included the width of the debris (denoted as *W*), its submerged depth (referred to as *T*), the upstream length of debris (designated as $L_u$), and the downstream extension of debris (represented as $L_d$), as illustrated in Figure 3. The width of debris (*W*) was anticipated to predominantly impact the lateral (transverse) extent of scour, with minimal influence on the maximum scour depth, as observed in previous studies [19]. To account for the lateral extent of scour while mitigating potential effects from flume sidewalls, a uniform value of 12 cm was assigned to the *W* for all debris shapes. Table 1 lists various parameters from laboratory studies. Some studies did not include debris, so some columns are left blank. This experimental study fits within the ranges of parameters chosen by other researchers. The proportional size of the flume vs. pier (*B*/*D*), width of flume vs. debris (*B*/*W*), duration of testing (*te*), and debris dimensions and location with respect to the pier, are within the ranges of other studies.

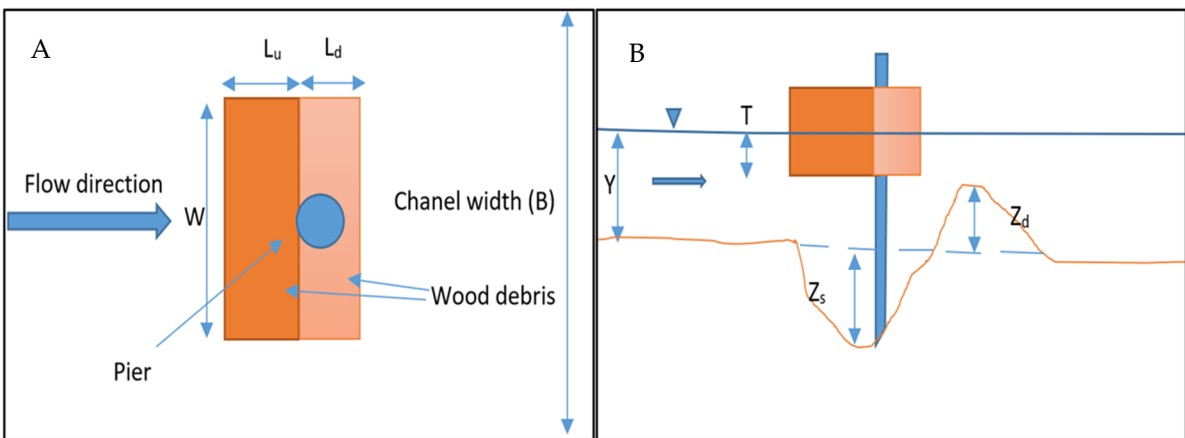

**Figure 3.** Key parameters ($W, T, L_u, L_d$) of floating debris illustrated in (**A**) top view and (**B**) side view.

**Table 1.** Hydraulic conditions and debris characteristics of the current and previous studies.

| Cases | B/D | B/W | te (min) | W/D | $L_u$/D | $L_u$/W | T/Y | $L_d$/$L_u$ |
|---|---|---|---|---|---|---|---|---|
| [2] | 9.0 | - | 220 | - | - | - | - | - |
| [5] | 9.8–14 | - | 100 | - | - | - | - | - |
| [13] | 6.7 | - | 2460–34,800 | - | - | - | - | - |
| [19,40] | 24 | 1.96 | 480–4320 | 6–24 | 3–24 | 0.5–1.5 | 0.3–1 | - |
| [21,22] | 8.3–20 | 1–7.7 | 360–5760 | 1.6–10 | 3 | 0.18–1.9 | 0.07–0.53 | 0–0.5 |
| [23] | 26 | 3.8–8.4 | 9000 | 3–7 | 3–7 | 1 | 0.09–1.95 | 1 |
| [35] | 6–7.5 | - | 120–180 | - | - | - | - | - |
| [36] | 12 | 2 | 300 | 0.015 | 0.32–3 | 0.05–0.5 | 0.2–0.4 | - |
| [41] | 10 | 2 | 360 | 5 | 0.3–2.5 | 0.06–0.50 | 0.07–0.18 | 0 |
| Current study | 15 | 2.5 | 360–1440 | 6 | 3–6 | 0.5–1 | 0.25–1 | 0–0.66 |

## 4. Testing Procedures

### 4.1. Preparations, Measurements, and Timing

Pre-test preparations required care to consistently build identical bed conditions and properly arrange the model pier and the *LWFD*. The sediment bed required precise smoothing around the pier and careful placement and compaction before each test. The channel-filling process continued slowly without creating disturbances until it reached the prescribed depth of 0.12 m. Adjusting the test discharge ($Q$) required a gradually increasing rate while observing and correcting any water level changes to maintain a constant level. The final step in establishing conditions included carefully establishing flow depth ($Y$) at steady-state conditions. Time began when the level of water matched the predetermined level ($Y$). Physical measurements for scour depth and extent used a point gauge, transparent scale, and non-contact laser scanning. The point gauge could be repositioned to measure several locations during the experiment with a precision of 0.01 mm. A transparent scale was also attached to the pier's side as a visual reference for pier scour depth. This method enabled continuous monitoring, which is especially useful in laboratory studies with clear water conditions. A non-contact laser scanning device was employed for continuous area measurements. It emitted a beam that reflected off the bottom of the scour hole and returned to the detector. This laser scanning approach produced a precise 3D contour of the scour hole, which later helped determine the scour hole area and volume (Figure 4). The scour depths on both sides of the pier were nearly equal due to symmetry. Therefore, one-sided measurements were taken.

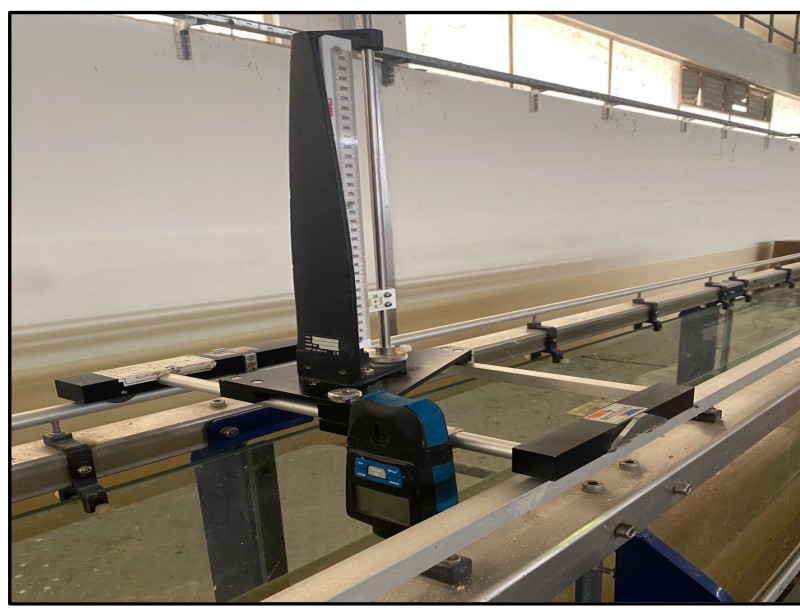

**Figure 4.** Point gauge and laser meter were positioned on the flume side walls during the measurements.

Scale measurements occurred at time = 1, 10, 20, 40, 80, 160, 240, and 360 min. After 6 h, analysis of the measurements showed that the changes in scour depth were less than 1 mm over the last 2 h. Because the growth rate was less than 3% of the pier width ($0.03D$) per hour [42], the 6 h duration test provided a reasonable standard comparison duration. In general, 50% to 80% of maximum scour may occur within 10% of the event duration [43]. As a check, some experiments extended 72 h to establish a baseline. As a result, the quasi-equilibrium conditions at 6 h proved adequate for the 110 experimental runs. The timeframe was long enough to complete at least 80–90% of the equilibrium scour. This approach significantly increased the quantity and variety of debris configurations while staying within the laboratory budget. Precise temporal effects may be overlooked in the current investigation since the goal was to examine the scour hole with and without debris collection after the same testing time. Following test completion, the water in the flume was drained entirely after the pump shut off. The point gauge and laser scanning devices performed measurements of the scour hole.

### 4.2. Testing Parameters Examined

The flume/pier/debris configurations totaled around 110 testing runs. A summary of debris combinations appears in Table 2. The top row lists the shapes, symbols, and debris positions shown in Figure 2. The percentage of obstruction ($A\%$) caused by the debris submerged frontal area ($A_d$) (normal to flow) was divided by the free channel cross-section: $A\% = (W - D)\,T/BY$, for a rectangular profile in the flow direction [37], and $A\% = (W - D)\,0.5T/BY$ for the triangle yield sign ($TY$). The total fontal area ($A_o$) is the entire area obstructed by debris buildup, as well as the pier. For rectangular debris, the profile is: $(A_o) = WT + D(Y - T)$, and for the triangle yield sign ($TY$), it is: $(A_o) = 0.5WT + D(Y - T)$. Conditions that remained consistent for every test were $W/B = 0.4$, $Y/D = 6$, and $V/Vc = 0.69$. Additional tests without debris supplemented the testing and provided baseline data.

**Table 2.** Testing matrix showing combinations of debris shapes, upstream and downstream positions, depth, and cross-section area.

| Symbol | $L_u$ /D $L_u$ (cm) | $L_d$/$L_u$ $L_d$ (cm) | T/Y T (cm) | A% | $A_d$ (cm$^2$) | $A_o$ (cm$^2$) |
|---|---|---|---|---|---|---|
| RB1 TB1HW1 LW1 TY1HC1 | 3 6.0 | 0 | 0.25 3.0 | 8.3 (3.47) * | 30 (15) | 54 (36.5) |
| RB2 TB2 HW2 LW2 TY2 HC2 | 3 6.0 | 0 | 0.5 6.0 | 16.7 (6.94) | 60 (30) | 84 (49) |
| RB3 TB3 HW3 LW3 TY3 HC3 | 3 6.0 | 0 | 1.0 12.0 | 33.3 (13.88) | 120 60) | 144 (74) |
| RB4 TB4 HW4 LW4 TY4 HC4 | 6 12.0 | 0 | 0.25 3.0 | 8.3 (3.47) * | 30 (15) | 54 (36.5) |
| RB5 TB5 HW5 LW5 TY5 HC5 | 6 12.0 | 0 | 0.5 6.0 | 16.7 (6.94) | 60 (30) | 84 (49) |
| RB6 TB6 HW6 LW6 TY6 HC6 | 6 12.0 | 0 | 1.0 12.0 | 33.3 (13.88) | 120 60) | 144 (74) |
| RB7 TB7 HW7 LW7 TY7 HC7 | 3 6.0 | 0.33 2.0 | 0.25 3.0 | 8.3 (3.47) * | 30 (15) | 54 (36.5) |
| RB8 TB8 HW8 LW8 TY8 HC8 | 3 6.0 | 0.33 2.0 | 0.5 6.0 | 16.7 (6.94) | 60 (30) | 84 (49) |
| RB9 TB9 HW9 LW9 TY9 HC9 | 3 6.0 | 0.33 2.0 | 1.0 12.0 | 33.3 (13.88) | 120 60) | 144 (74) |
| RB10 TB10 HW10 LW10 TY10 HC10 | 6 12.0 | 0.33 4.0 | 0.25 3.0 | 8.3 (3.47) * | 30 (15) | 54 (36.5) |
| RB11 TB11 HW11 LW11 TY11 HC11 | 6 12.0 | 0.33 4.0 | 0.5 6.0 | 16.7 (6.94) | 60 (30) | 84 (49) |
| RB12 TB12 HW12 LW12 TY12 HC12 | 6 12.0 | 0.33 4.0 | 1.0 12.0 | 33.3 (13.88) | 120 60) | 144 (74) |
| RB13 TB13 HW13 LW13 TY13 HC13 | 3 6.0 | 0.66 4 | 0.25 3.0 | 8.3 (3.47) * | 30 (15) | 54 (36.5) |
| RB14 TB14 HW14 LW14 TY14 HC14 | 3 6.0 | 0.66 4 | 0.5 6.0 | 16.7 (6.94) | 60 (30) | 84 (49) |
| RB15 TB15 HW15 LW15 TY15 HC15 | 3 6.0 | 0.66 4 | 1.0 12.0 | 33.3 (13.88) | 120 60) | 144 (74) |
| RB16 TB16 HW16 LW16 TY16 HC16 | 6 12.0 | 0.66 8.0 | 0.25 3.0 | 8.3 (3.47) * | 30 (15) | 54 (36.5) |
| RB17 TB17 HW17 LW17 TY17 HC17 | 6 12.0 | 0.66 8.0 | 0.5 6.0 | 16.7 (6.94) | 60 (30) | 84 (49) |
| RB18 TB18 HW18 LW18 TY18 HC18 | 6 12.0 | 0.66 8.0 | 1.0 12.0 | 33.3 (13.88) | 120 60) | 144 (74) |

* The areas for *TY* tests were smaller due to their immersed shape.

*4.3. Scour Evolution during the Tests*

It is crucial to note that the occurrence of the maximum scour surface and volume did not always coincide with the time of maximum scour depth because local sediment would fill the scour hole due to collapse or upstream sources. Observations revealed that downflow directly in front of the pier caused the scour depth to rapidly reach its maximum. The scour zone then continually evolved, briefly, with neighboring sand slipping into the scour hole and moving downstream [35]. As a result, the maximum scouring area/volume often occurred later than the maximum scouring depth. For simplicity reasons, we assumed

that once the maximum scour depth occurred, the maximum scouring area or volume also occurred.

In single-pier experiments without debris, a vertical pressure gradient developed along the face of the pier as flow impacted it. The gradient produced a downflow jet that agitated the sediment bed and removed the sand. The main vortex that developed around the pier removed a significant portion of the bed materials and created the scour hole. Wake vortices formed by the flow splitting at the pier corners swept the materials from the scour hole and transported them downstream. The presence of the *LWFD* at the bridge pier greatly influenced the area's scouring characteristics and flow pattern. For experiments with the *LWFD* accumulation, Figure 5 illustrates the scour hole shape for a baseline test after the water was drained from the flume. Measurements of debris collection revealed varying scour levels and sand deposit geometries depending on the hydraulic conditions and morphological debris parameters.

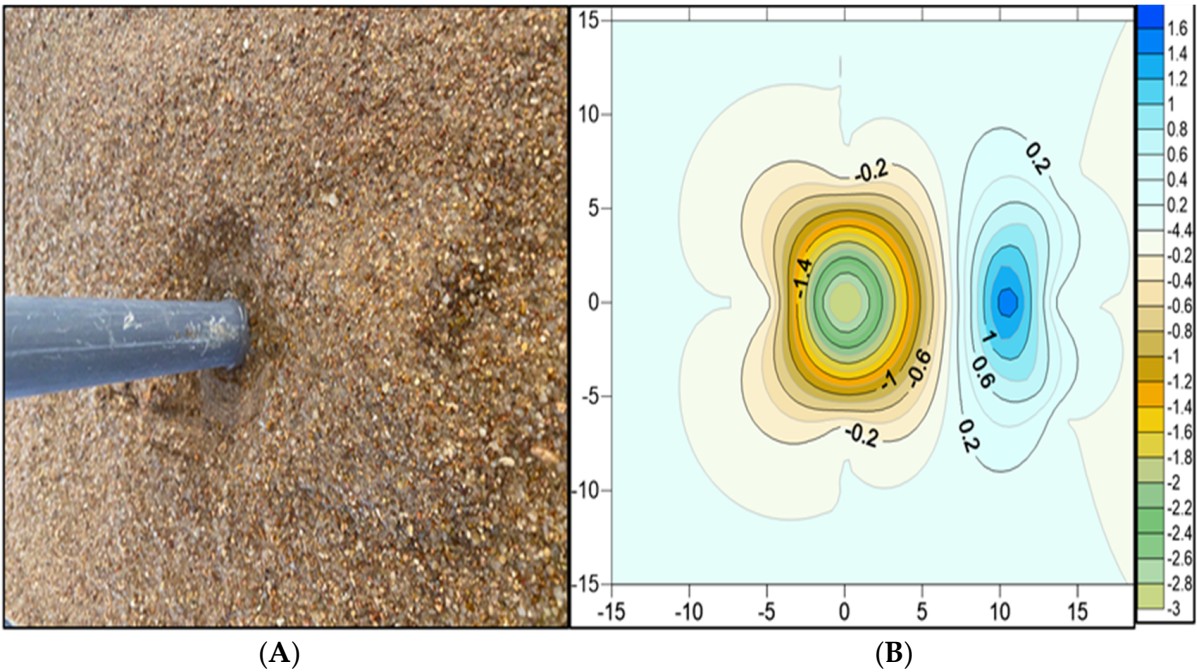

**(A)**                                   **(B)**

**Figure 5.** Scour hole characteristics around a pier in no−debris conditions: (**A**) scour hole configuration after water was drained out, and (**B**) contour map highlighting maximum scour and deposition depth. All numbers are in centimeters.

## 5. Test Results and Discussion

### 5.1. Maximum Scour Depth and Deposition Height for Different Debris Shapes

This research investigated six debris shapes. The conditions listed in Table 2 summarize the various parameters studied in this investigation. The parameters included debris shape, debris position concerning the pier ($L_d/L_u$), submerged depth ($T$), and the resultant degree of channel obstruction due to its immersed cross-section ($A\%$). Debris accumulation obstruction percentages ($A\%$) were varied due to changes in the debris' submerged depth ($T$), ranging from low to high, 8.3% to 33.3%, and additionally from 3.47% to 13.9% for the triangle yield ($TY$) case only.

Figure 6 summarizes the relative impact of the debris configurations. The vertical axis represents the ratio of maximum scour depth to the pier width ($Zs/D$). Each group of vertical bars comes from a different debris shape. Each color corresponds to a specific immersion depth ($T$ = 3, 6, and 12 cm) and distance upstream from the pier ($L_u$ = 6 and 12 cm). The high wedge shapes $HW3$ and $HW6$ produced the deepest scour (($Zs/D$) = 3.8 and 4.1, respectively). The high wedge shape also produced significant scour for all

conditions ($HW1 - HW6$). The triangle yield ($TY1$ and $TY6$) was the least impactful shape, which produced less scour.

**Figure 6.** Variation of relative scour depth ($Zs/D$) with different debris shapes, sizes, and depths when $L_d/L_u = 0$.

Figures 7 and 8 illustrate the profiles of scour holes and the sand deposition height near the pier location, showing the longitudinal (*X*-axis), transverse (*Y*-axis), and vertical (*Z*-axis) dimensions. Various factors, including the upstream debris length ($L_u$), obstruction percentage ($A\%$), and debris shape, significantly influenced the morphology of scour holes. When the $A\%$ was high and ($L_u$) was 6 cm, the scour holes tended to be deeper and steeper, and the sand was deposited near the pier location. This effect intensified with increasing $A\%$, leading to enhanced flow redirection and steeper upstream slopes. However, when ($L_u$) was 12 cm, irrespective of the $A\%$ value, the scour hole tended to exhibit milder and shallower characteristics, except for the high wedge configuration.

Notably, the high wedge configuration ($HW6$) consistently yielded larger scour hole volumes, areas, and higher deposition heights ($Z/D$) than other debris shapes, underscoring its notable impact on scour hole morphology.

Figures 6–8 illustrate several general trends regarding the impact of different debris shapes on pier scour:

1. All debris with shallow immersion depths ($T$ = 3 cm, blue and yellow lines) produced less scour than with deeper immersion: $T$ = 6 and 12 cm.
2. Pier scouring increased significantly when the obstruction percentage ($A\%$) was high.
3. Debris shapes positioned further upstream ($L_u$ = 12 cm, second group—yellow, blue, and green) generally caused less scour compared to those when $L_u$ = 6 cm (first group), regardless of the increase in the ($A\%$) value. However, the high wedge ($HW$) showed an increased scour depth with a longer upstream debris length at $L_u$ = 12 cm.
4. Regardless of shape, the deepest scours occurred when the debris was at full depth ($T$ = 12 cm higher obstruction ratio ($A\%$)) and near the pier ($L_u$ = 6 cm), shown by gray lines.

Based on the observations, it is evident that bridge piers configured with the high wedge ($HW$) shape exhibited scour holes deeper than other shapes for all scenarios. For HW conditions, the flow was redirected downward toward the channel bed immediately adjacent to the pier face by the inclined debris surface. This would drive the apex of the horseshoe vortex even harder. The $HW$ configuration led to the worst-case scour condition, with a maximum scour depth of around 70% more than the no−debris case. This finding is consistent with findings in existing literature [19,41,44]. Contrarily, this trend was not

evident for scenarios with $L_u$ = 12 cm in other debris shapes. In these instances, the elongation of debris upstream led to diminished flow velocities surrounding the piers (*HW* at its base is in the same place with $L_u$ = 6 or 12 cm). This reduction contributed to the dissipation of vortex power and decreased bed shear, reducing the scouring effect on the pier. Therefore, despite the rise in the obstruction ratio (*A*%), the scour depth remained relatively shallow.

The presence of rectangular (*RB*) debris significantly impacted pier scour dynamics. Unlike the classic "horseshoe vortex" pattern observed at piers without debris, flow at a pier with rectangular debris was substantially obstructed. Instead of spiraling past the pier, the flow was forced to plunge beneath the upstream face of the debris. This plunging flow created a consistent upstream scour trough. The blocky masses characteristic of the rectangular (*RB*) debris shape induced scour hole development by causing a pronounced deviation of current lines when encountering the pier. This deviation led to augmented flow separation and the formation of larger wake vortexes. Figure 9 offers a three-dimensional representation of how rectangular debris influenced scour behavior.

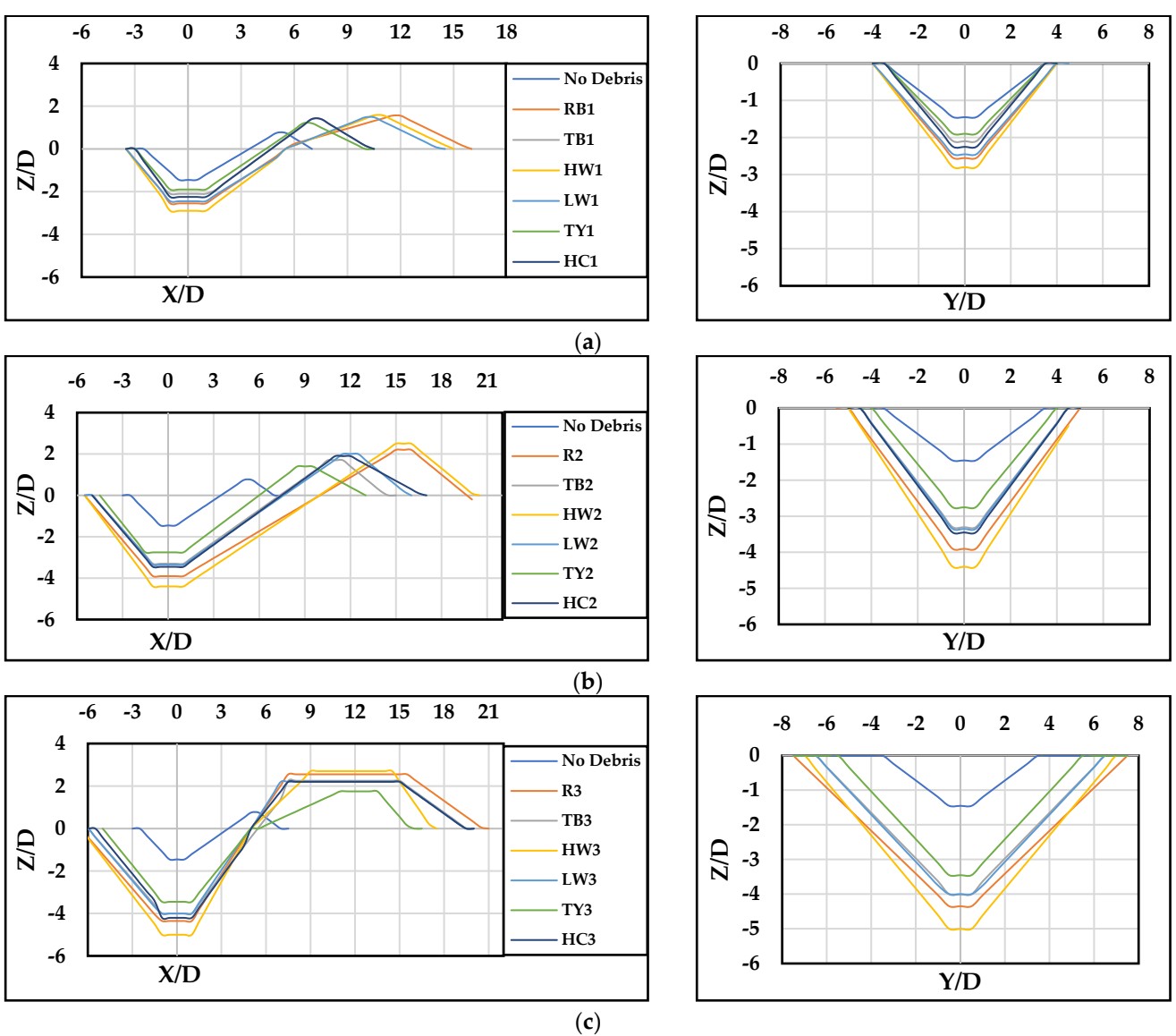

**Figure 7.** The scour longitudinal and transverse profiles when $L_d/L_u$ = 0, with different thicknesses (*T*), widths (*W*), and upstream lengths (*$L_u$*) configurations: (**a**) 3−12−6, (**b**) 6−12−6, and (**c**) 12−12−6.

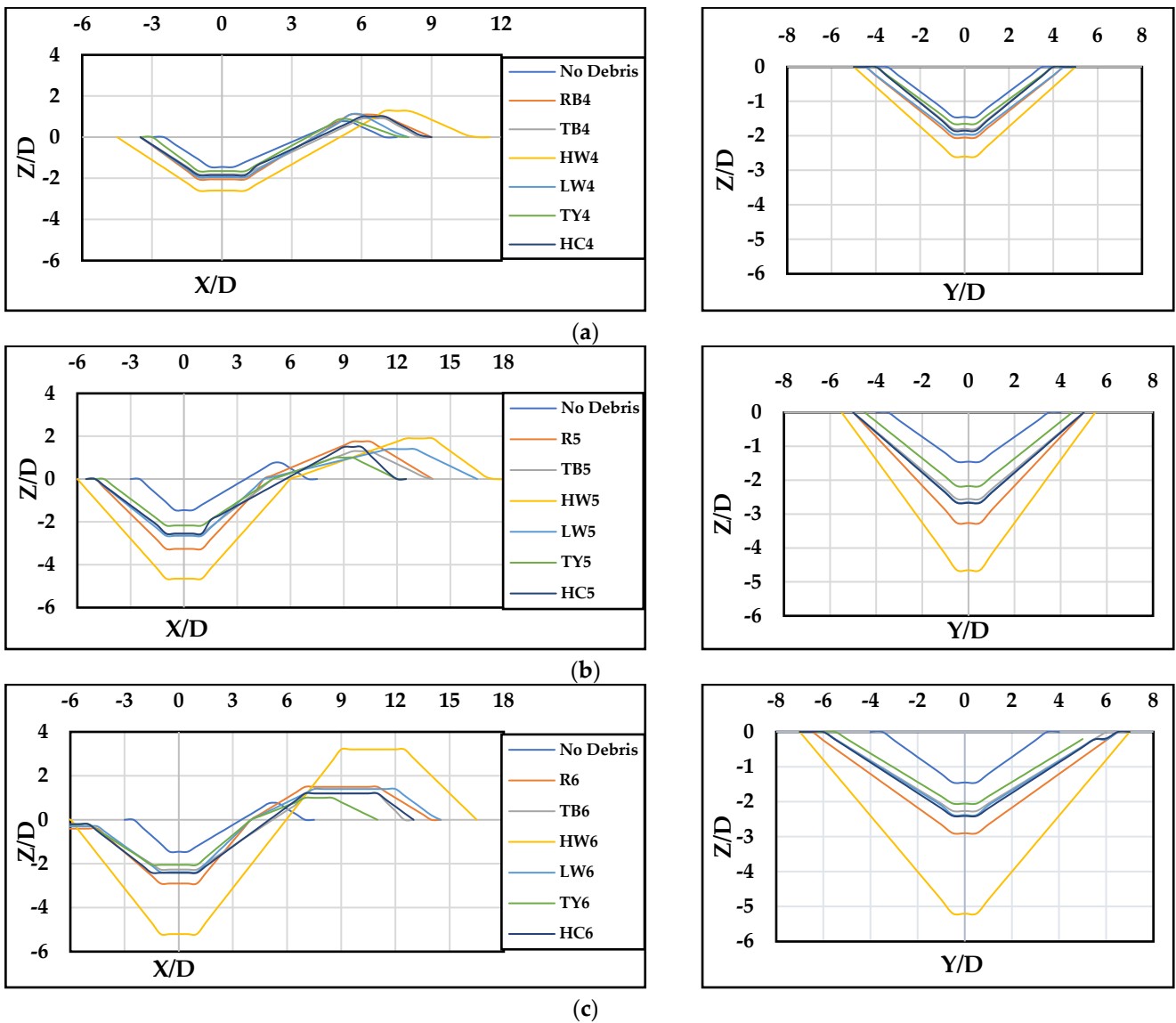

**Figure 8.** The scour longitudinal and transverse profiles when $L_d/L_u = 0$, with different submergence depth ($T$), width ($W$), and upstream length ($L_u$) configurations: (**a**) 3−12−12, (**b**) 6−12−12, and (**c**) 12−12−12.

Cases involving the triangle bow (*TB*) and half-cylinder (*HC*) shapes displayed flow separation tendencies, likely due to their streamlined configurations. This streamlined shape reduces the disruption of flow patterns, resulting in less pronounced scour at the pier face compared to other debris shapes (similar to the delta vane that is attached to the cylindrical pier and can minimize shear stress on the bed, which is approximately 30% less than the cylindrical pier alone [45]). Furthermore, the extent of scour at the pier face is closely linked to the thickness of the debris blockage. A greater thickness of debris directly lodged against the pier tends to create a more significant scour at the pier face. This phenomenon is particularly evident in triangle debris shapes, where the geometry of the debris exacerbates the scouring effects at the pier face.

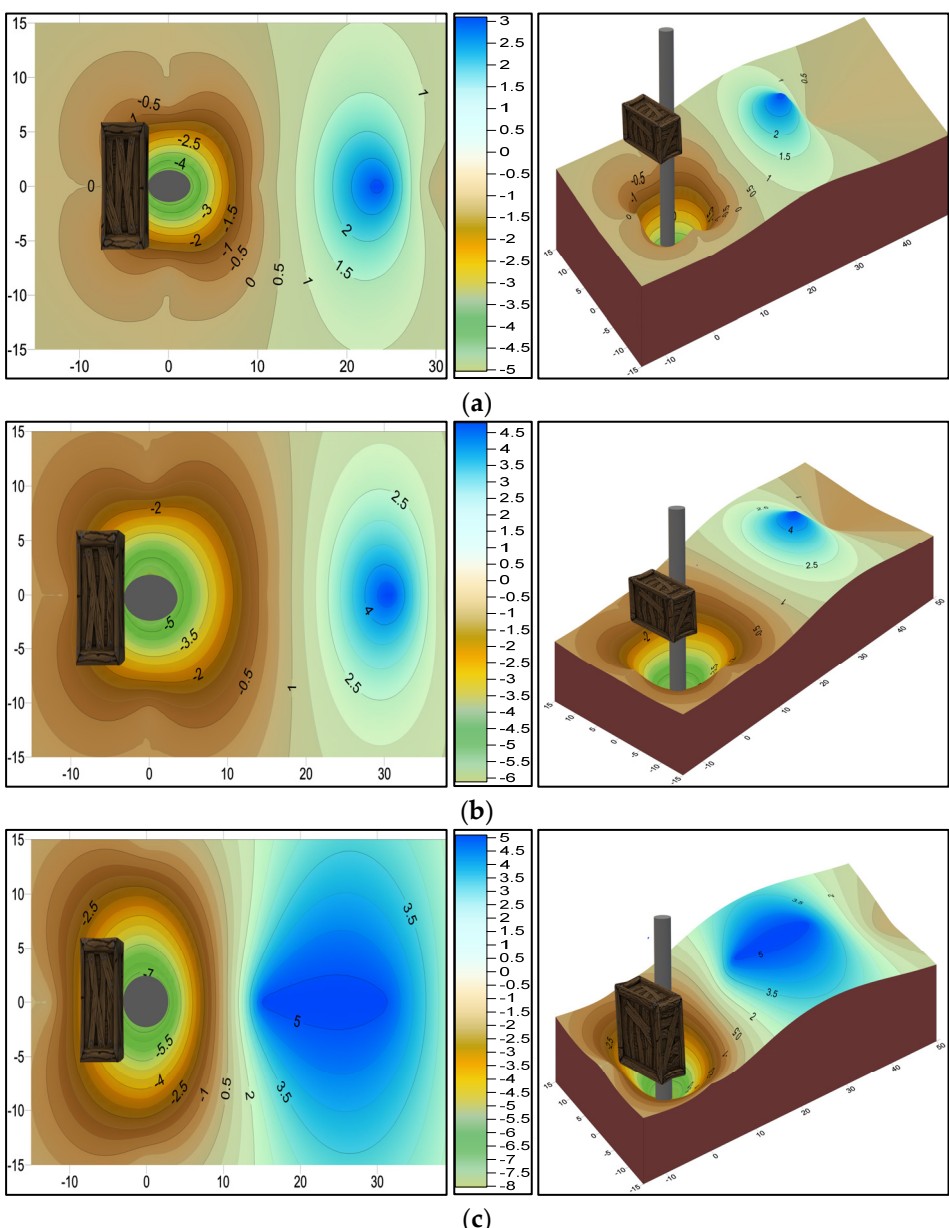

**Figure 9.** Contour map and 3D surface layer of scour profiles for rectangular debris ($RB1 - RB3$) with different submergence depth ($T$), width ($W$), and upstream length ($L_u$) configurations: (**a**) $3-12-6$, (**b**) $6-12-6$, and (**c**) $12-12-6$.

The scour depth associated with the low wedge (*LW*) shape exhibited similar behavior to the rectangle block (*RB*) debris, albeit with lower scour depth values. This can be attributed to the upward inclination shape of the low wedge configuration, which redirects the flow in an upward direction. As a result, the scour depth tended to be reduced, particularly for debris with lower thickness (low *A*%). Interestingly, when the thickness (*T*) of the debris reached 12 cm, high *A*%, the scour depth associated with the low wedge shape approached that of the rectangle debris. The interaction between the unique geometric characteristics of the low wedge shape and the flow dynamics can explain this phenomenon. Despite the upward redirection of flow, the increased debris thickness contributes to more significant obstruction, thereby intensifying the scouring effects at the pier face.

The triangle yield (*TY*) debris shape exhibited distinct characteristics that contribute to shallower scour depths compared to other shapes. Firstly, its cone profile produced a lower obstruction ratio (*A*%) even with a high debris thickness, allowing more flow to pass

beneath and around it. Secondly, the *TY* shape increased local flow velocity beneath it while confining it to a smaller region, minimizing disruption to nearby flow patterns and reducing scour potential. Additionally, the angled design of the *TY* shape directed flow along its surface toward the streambed in a streamlined manner, mitigating flow separation and the formation of large horseshoe vortices, which are known contributors to scouring.

Overall, the total scour at the pier experienced a notable increase when the *A%* was substantial. Under these conditions, pressure flow and contraction effects resembled those induced by debris, such as the pressure flow observed beneath bridge decks submerged during floods [19].

*5.2. Debris Downstream Extension ($L_d/L_u$)*

Figure 10 illustrates the relative scour depth results ($Zs/D$) for debris cases where $L_d/L_u$ = 0.33, including variations in the downstream debris length and distance upstream from the pier ($L_u$ = 6 and 12 cm). Debris accumulation obstruction percentages (*A%*) ranged from 8.3% to 33.3%, and from 3.47% to 13.9% for the triangle yield (*TY*) case only.

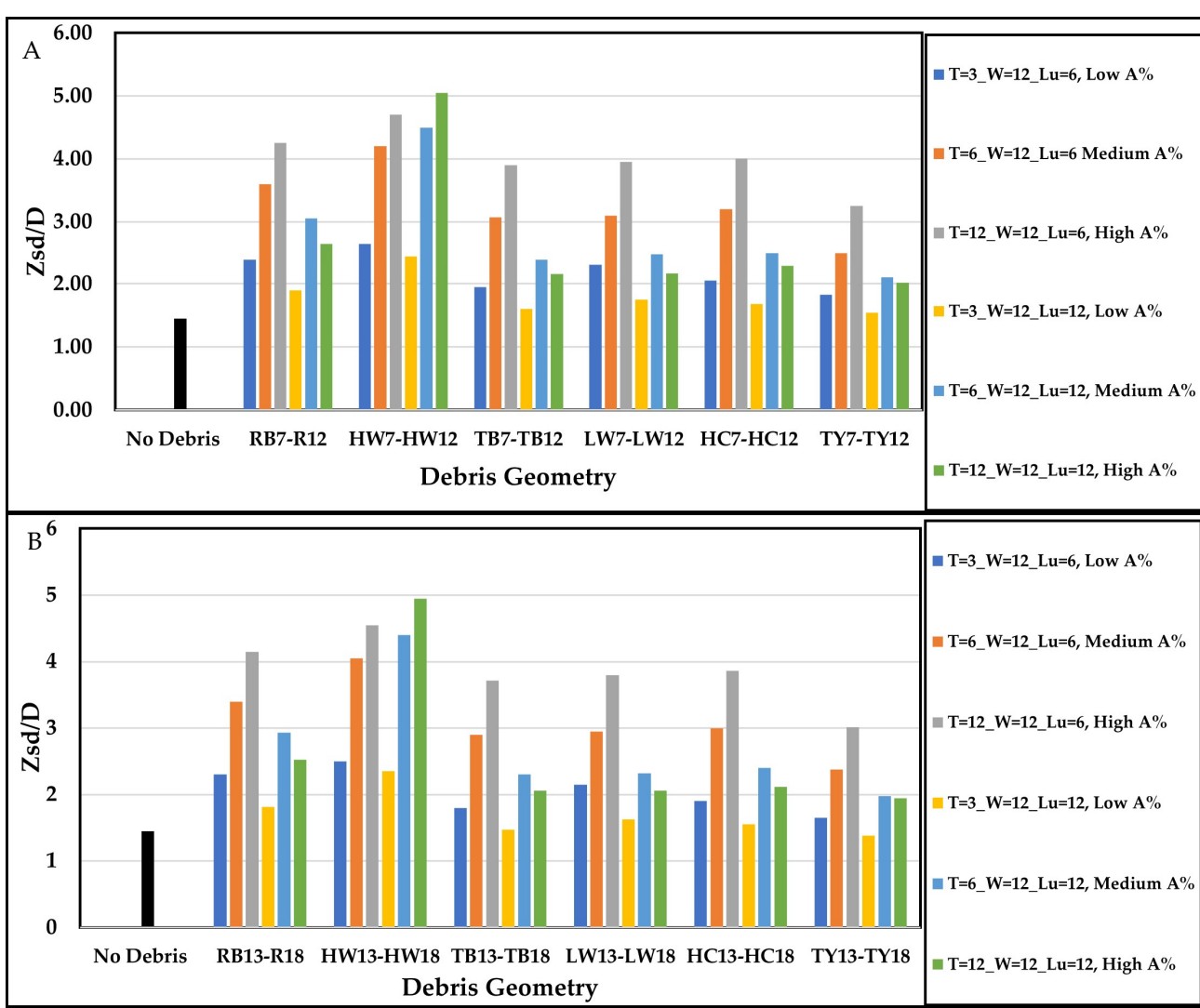

**Figure 10.** Variation of relative scour depth ($Zsd/D$) with different debris shapes, sizes, and positions for (**A**) $L_d/L_u$ = 0.33 and (**B**) $L_d/L_u$ = 0.66.

In general, the results showed that all parameter ranges exhibited consistent behavior, similar to scenarios where debris accumulates solely on the upstream side of the pier ($L_d/L_u$ = 0). However, under identical hydraulic conditions and specific *LWFD* geometries,

an increase in the $L_d/L_u$ ratio to 0.33, particularly at maximum $A\%$ (the debris at full depth), signified the downstream movement of the debris, resulting in a scour depth reduction but an increase in scour hole dimensions, as indicated in Figure 11.

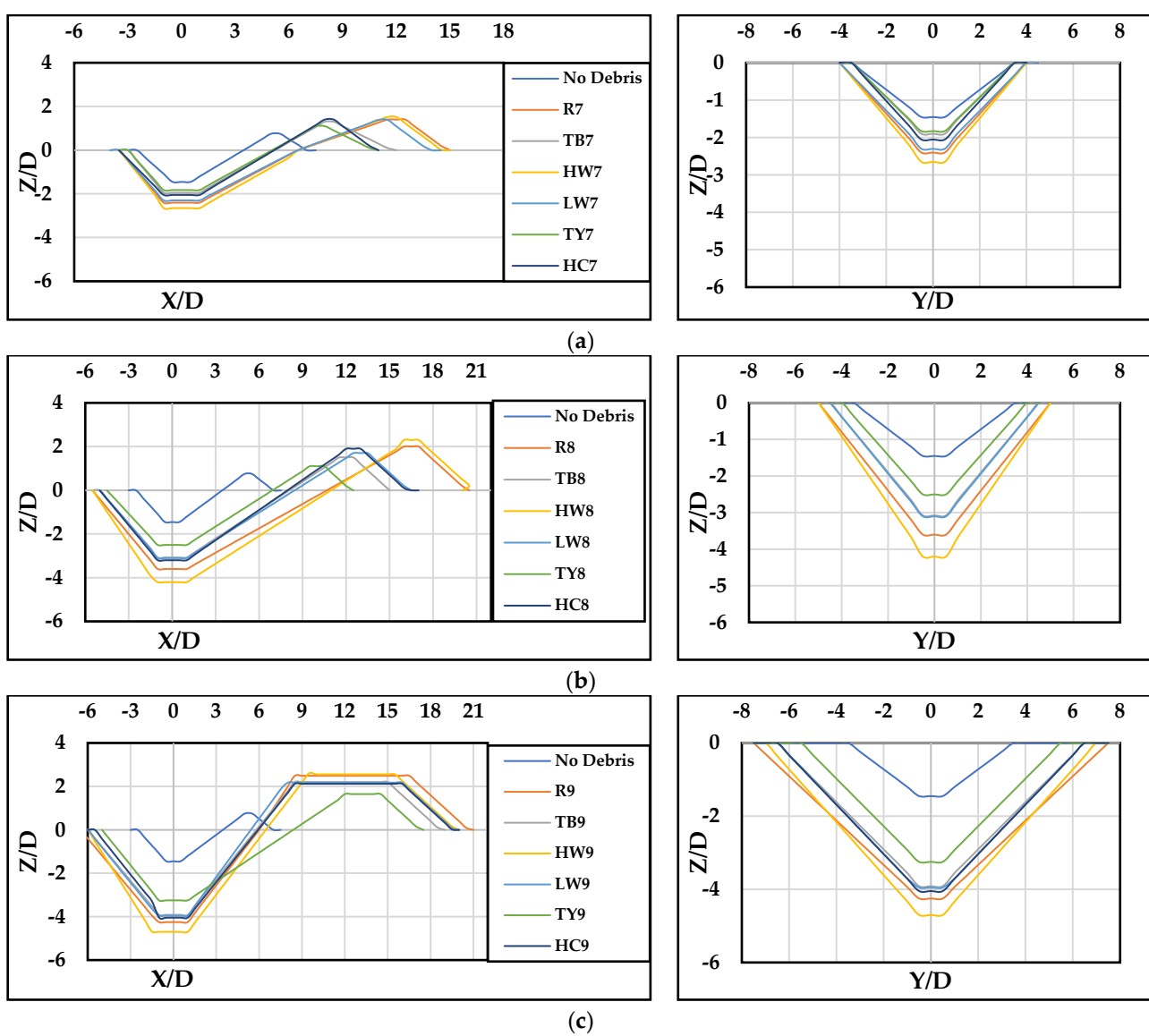

**Figure 11.** The scour longitudinal and transverse profiles when $L_d/L_u$ = 0.33, with different thicknesses (*T*), widths (*W*), and upstream lengths ($L_u$) configurations: (**a**) 3−12−6, (**b**) 6−12−6, and (**c**) 12−12−6.

During the tests of downstream debris extension scenarios, the movement of bed material downstream of the pier was first seen because of wake vortices that formed behind the debris expansion block. These vortices carried bed particles into the main flow and downstream, where they eventually deposited near the pier downstream as a dune. The formation of the horseshoe vortex, in addition to the existence of the wake vortex, marked the onset of scour hole development. The horseshoe vortex developed upstream of the pier and then on both sides equally, leading to a rapid increase in scour depth. As the scour depth increased, the power of the horseshoe vortex decreased, resulting in a reduction in the rate of scour hole expansion over time until quasi-equilibrium was achieved after around 5 h. The alteration of vortex power in the pier scour hole involves increasing and decreasing tendencies with the growth of the scour hole [6].

Downstream debris extension can weaken the vortex group. This weakening, corroborated by the rapidly diminished horseshoe vortices in this study, potentially deflects the upstream flow over a large area around the pier due to downstream movement of the debris and, consequently, a reduced scour depth directly below the pier. This observation aligns with previous research [22]. However, a seemingly contradictory aspect lies in the increased scour hole area with the extension of downstream debris. The increased hole area suggests a spreading effect on the scour pattern despite the overall reduction in scour depth. The debris' role in the flow redirection can explain this phenomenon. As the flow deflected downstream, it interacted with a larger streambed area, which eroded laterally to accommodate the displaced flow.

Figure 12 shows the relative scour depth and maximum deposition height in both longitudinal and transverse slices while setting ($L_d/L_u$) to 0.66. Compared to the previous measurements ($L_d/L_u = 0.33$), the scour depth consistently decreased across all parameter ranges that were studied. This decrease became more noticeable as the blockage percentage ($A\%$) rose. Notably, the longitudinal profiles revealed a downstream shift in the location of maximum deposition for most cases, expanding the scour hole area and slightly increasing the transverse length versus the longitudinal direction. These observations suggest that as the debris extended further downstream, a notable reduction in scour depth occurred, possibly due to increased flow redirection and a wider area of scour hole development.

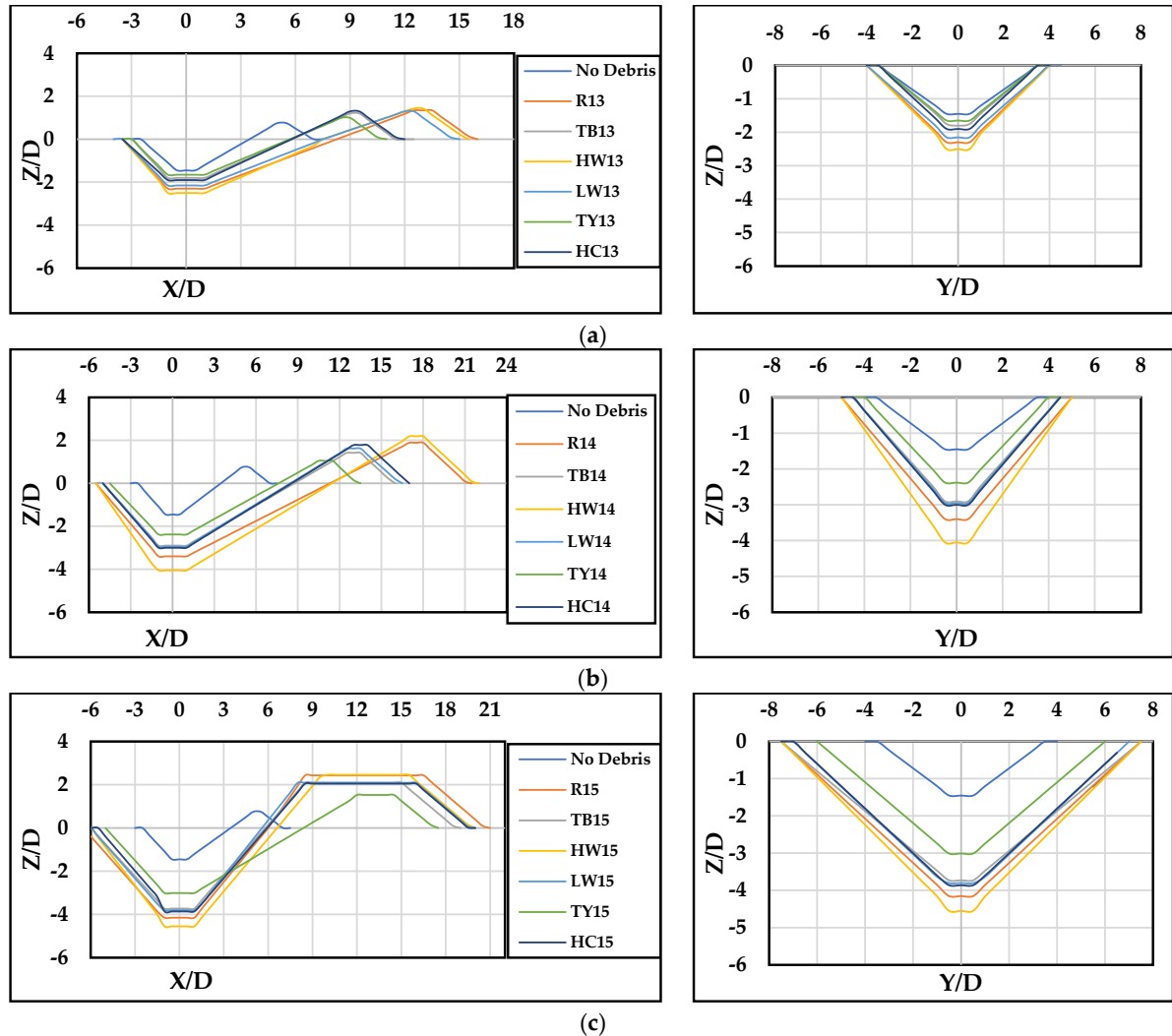

**Figure 12.** The scour longitudinal and transverse profiles when $L_d/L_u = 0.66$, with different thicknesses ($T$), widths ($W$), and upstream lengths ($L_u$) configurations: (**a**) 3−12−6, (**b**) 6−12−6, and (**c**) 12−12−6.

Figure 13A,B describe the influence of increasing the debris extension ratio ($L_d/L_u$) from 0 to 0.66 on scour depth ($Zs/D$) for various debris shapes and $A\%$ values when the debris upstream length was $L_u$ = 6 cm and 12 cm, respectively. Two figures were examined, both depicting $Zs/D$ on the $Y$-axis and $L_d/L_u$ on the x-axis. both figures demonstrate a general decrease in $Zs/D$ with increasing $L_d/L_u$ for all scenarios. The analysis suggested that a larger upstream debris length ($L_u$ = 12 cm) might lead to a more significant reduction in scour depth for a given increase in $L_d/L_u$. The altered flow patterns brought about by the presence of debris can be responsible for the observed decrease in scour depth with increasing $L_d/L_u$. As $L_d/L_u$ increased, there was a notable trend across all debris shapes: the scour depth decreased. This was particularly evident with shapes such as rectangular blocks (*RB*) and *LW*, which created substantial obstruction to flow due to their solid and large surface areas, resulting in pronounced reductions in scour depth. Conversely, shapes resembling *HW* may allow water to bypass more easily due to their streamlined profiles, leading to comparatively less severe reductions in scour depth. Shapes such as *HC* or *TB* exhibited intermediate characteristics, with moderate reductions in scour depth due to their rounded or tapered shapes. Debris shapes resembling *TY* offered complex flow patterns, and the degree of obstruction and subsequent scour depth reduction could vary based on factors such as branch arrangement and angle.

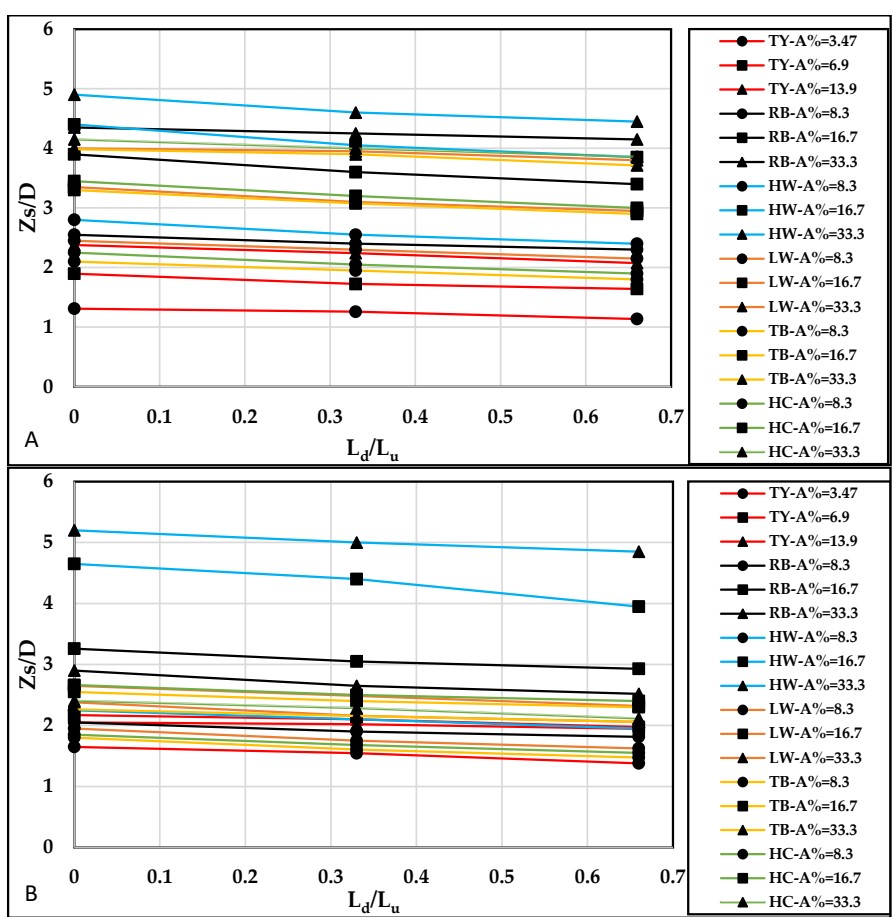

**Figure 13.** The relative scour depth ($Zs/D$) vs. the ($L_d/L_u$) range 0 to 0.66 for all tested data when (**A**) $L_u$ = 6 cm and (**B**) $L_u$ = 12 cm.

This hypothesis suggests that if the debris scenarios depicted in both figures are similar, it implies that debris with a greater extent upstream can offer increased protection against scouring around piers. However, this heightened protective effect may coincide with a concomitant increase in the positional force exerted on the pier.

## 6. The Volume and Area of the Scouring Hole

The dimensions of the pier scour hole directly correlate with the amount of eroded soil surrounding the pier structure. Figure 14 presents various scour map scenarios, including no−debris-induced and rectangular (*RB*) debris-induced, with different scenarios and downstream debris extension ratios ($L_d/L_u$) of 0, 0.33, and 0.66 (*RB*1 − 3, *RB*7 − 9, and *RB*13 − 15). For these tests, the scouring area and volume development typically lagged in attaining the final scour depth. However, for simplification purposes, this study assumed both conditions occur simultaneously. Therefore, a one-sided scour hole area and volume were calculated. The expressions for a differential area or volume were integrated to determine the total area and total volume, as shown in the equations below:

$$The\ area\ of\ scour\ hole\ (A) = \int_{x0}^{xn} y\ dx \tag{1}$$

$$The\ volume\ of\ scour\ (V) = \int_{x0}^{xn} Z_s A(x,y)dx \tag{2}$$

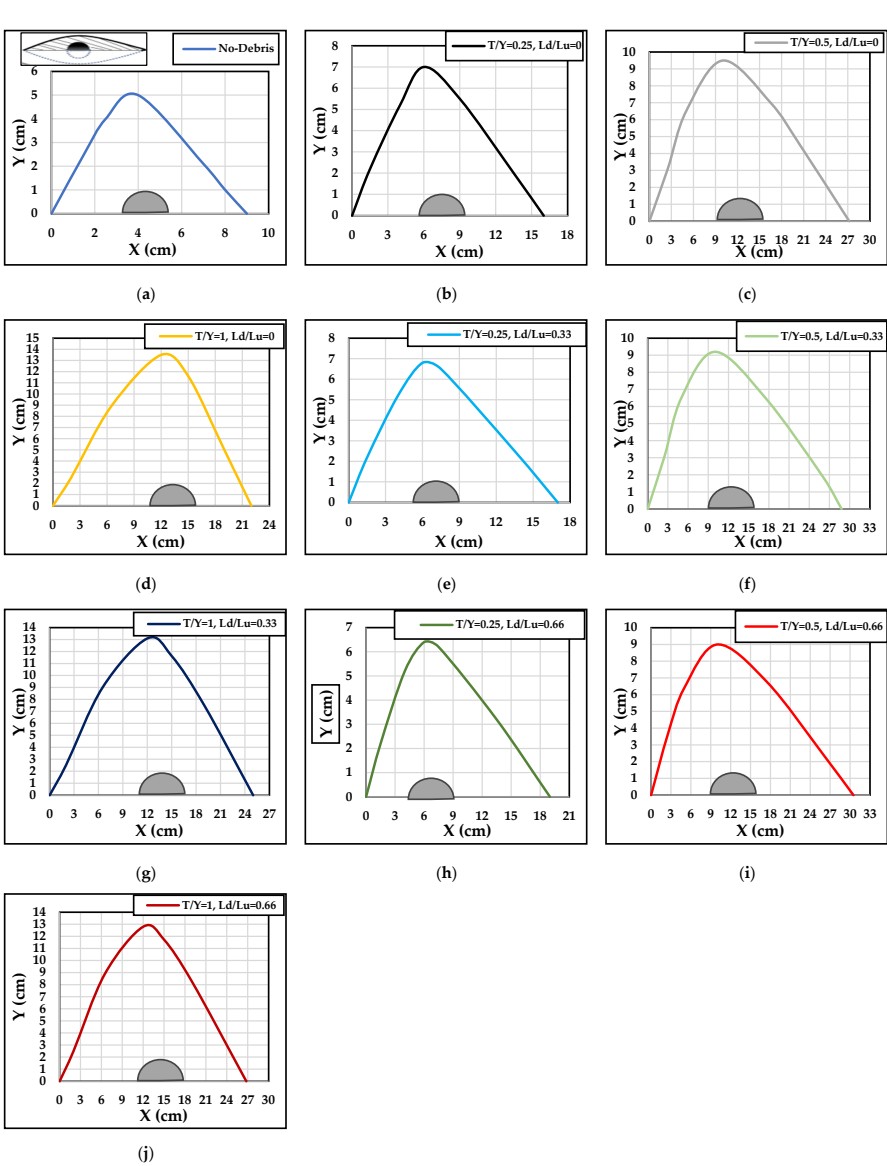

**Figure 14.** Schematic distribution of the scour hole and dimensions along the $(X − Y)$ axis for various scenarios: (**a**) no debris, (**b**) *RB*1, (**c**) *RB*2, (**d**) *RB*3, (**e**) *RB*7, (**f**) *RB*8, (**g**) *RB*9, (**h**) *RB*13, (**i**) *RB*14, and (**j**) *RB*15.

In the context of debris presence, particularly rectangular (*RB*) debris, significant alterations in the scour hole dimensions were observed, as illustrated in Figure 15A. With no debris, the scour hole area (*A*), using Equation (1), measured around 50 cm$^2$. However, when rectangular debris was introduced with no downstream extension ($L_d/L_u = 0$), the scour hole area expanded considerably to 128 cm$^2$, 300 cm$^2$, and 351 cm$^2$ for debris thicknesses (*T*) of 3 cm, 6 cm, and 12 cm, respectively. The debris substantially increased the scour area due to the obstruction it caused.

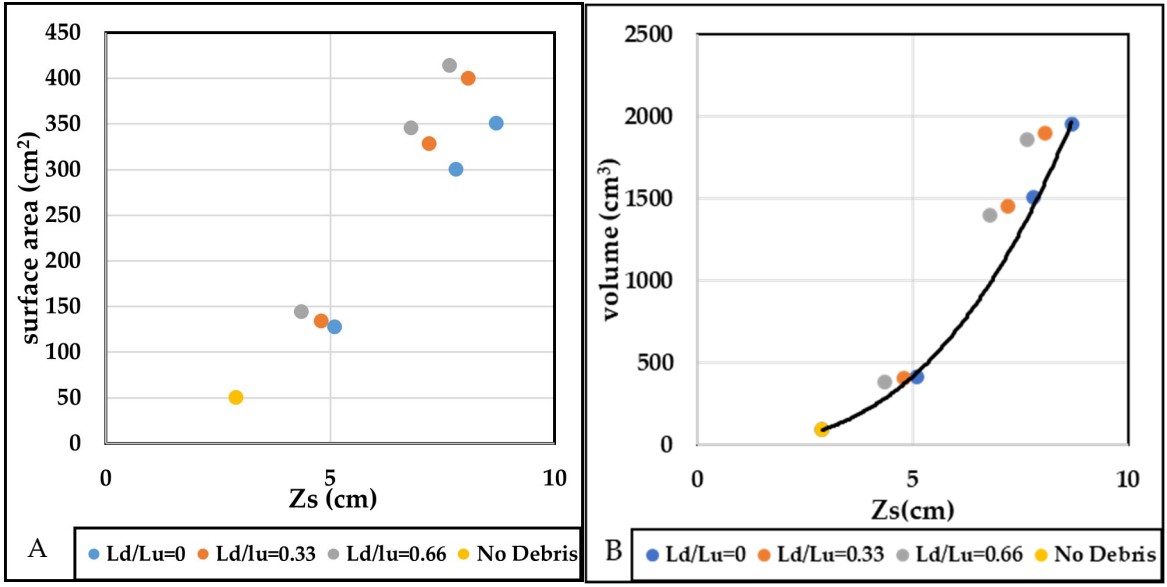

**Figure 15.** (**A**) Calculation of the scour hole area across various scenarios employing Equation (1). (**B**) Determination of the scour hole volume across different scenarios utilizing Equation (2).

Interestingly, when a downstream extension of the debris occurred ($L_d/L_u = 0.33$ and $L_d/L_u = 0.66$), the area still increased compared to the no downstream debris condition ($L_d/L_u = 0$), albeit at a lower rate. For instance, at $L_d/L_u = 0.33$, the area increased by 4.7%, 9.3%, and 14% for debris thicknesses (*T*) of 3 cm, 6 cm, and 12 cm, respectively, compared to $L_d/L_u = 0$. Similarly, at $L_d/L_u = 0.66$, the area increased by 12.5%, 15.3%, and 18% for the corresponding debris thicknesses.

However, the scour hole volume (*V*, Equation (2)) exhibited a different trend. Despite the increase in area, the scour hole volume experienced a reduction when downstream debris was present ($L_d/L_u \neq 0$). At $L_d/L_u = 0.33$ and 0.66, the scour volume decreased compared to the no−debris scenario, indicating that although the area increased, the overall excavation of the scour hole was shallower, as illustrated in Figure 15B. Additionally, the volume (*V*) of the scour hole was correlated with the scour depth (*Zs*) through this equation $V = 4.5Zs^{2.81}$. This relationship highlights the direct proportionality between scour depth and scour hole volume, suggesting that as the scour depth increased, the volume of material excavated also increased consistently.

## 7. Predicting Debris Scour Depth with Empirical Equations

Several scour depth prediction equations were chosen to compare with the experimental results of this investigation. The proposed approaches commonly applied are:

1. Melville and Sutherland (1988) [27]
2. Richardson and Davies (CSU) (2001) [46]
3. May, Ackers, and Kirby [47]
4. Sheppard et al. (2014) [33]

The original equations were initially devised to calculate scour depth in scenarios featuring isolated piers without considering debris. To address this limitation, researchers

integrated the concept of comparable pier width, as introduced in [23] (Equation (3)) and [19] (Equation (4)), to analyze the impact of debris−induced scour.

Following Melville and Dongol [23], the accumulation of debris enlarges the effective width of a pier, thereby constraining flow and amplifying the streamwise and downward kinetic energy close to the pier. In scenarios involving debris accumulation, designers frequently utilize an effective width derived from the dimensions of both the pier and the debris [23]. Melville's equation, outlined below, serves as a fundamental method for calculating this effective width, denoted as *De*, at bridge piers with debris accumulations:

$$De = \frac{T^*W + (Y - T^*)D}{Y} \tag{3}$$

where, $T^* = 0.52^*$, $T$ and $W$ are the debris submerged thickness and width, $Y$ is the flow depth, and $D$ is the pier width.

Building upon the foundation laid in [23], Lagasse et al. [19] proposed an innovative equation to address the effects of rectangular and triangular debris accumulating against square piers for calculating the equivalent pier width (*a*). This equation, presented below, takes into account the significant influence of debris thickness, *T*, on the downflow intensity experienced at the pier:

$$a = \frac{Kd1 \times TW \times \left(\frac{L}{Y}\right)^{Kd2} + (Y - Kd1 \times T)D}{Y} \tag{4}$$

where $Kd1 = 0.39$ and $0.14$ are the rectangular and triangular debris shape factors. $Kd2 = -0.79$ and $-0.17$ are the plunging flow intensity factors for the rectangular and triangular debris.

When comparing the calculated effective pier width (*De*) to the equivalent width (*a*), the effective pier width (*De*) tended to be an overestimate of the pier width, especially when the debris was shaped as triangles. The overestimation was a result of the shape of the debris and its upstream extension length ($L_u$), not accounting for the *De* in Equation (3). The effective and equivalent width equations were proposed and rigorously validated using data from this study (Tables 3 and 4). The evaluation primarily centered on situations where debris was minimally submerged beneath the free surface, specifically encompassing both rectangular (*RB*) and triangle yield (*TY*) debris configurations.

**Table 3.** The effective pier width using Equation (3) and the calculated scour depth (*Z* − *CAL*).

| Cases (W∗$L_u$∗T) cm | Z−EXP (cm) | De (cm), Equation (1) | Z−CAL (cm) [27] | Z−CAL (cm) [43] | Z−CAL (cm) [46] | Z−CAL (cm) [47] | Z−CAL (cm) [33] |
|---|---|---|---|---|---|---|---|
| *RB* − 12 ∗ 6 ∗ 3 | 5.1 | 3.30 | 7.92 | 5.2 | 5.22 | 3.96 | 6.36 |
| *RB* − 12 ∗ 6 ∗ 6 | 7.8 | 4.60 | 11 | 7.1 | 6.5 | 5.4 | 8.58 |
| *RB* − 12 ∗ 12 ∗ 3 | 4.1 | 3.30 | 7.92 | 5.2 | 5.22 | 3.96 | 6.36 |
| *RB* − 12 ∗ 12 ∗ 6 | 6.52 | 4.60 | 11 | 7.1 | 6.5 | 5.4 | 8.58 |
| *TB* − 12 ∗ 6 ∗ 3 | 3.8 | 3.30 | 7.92 | 5.2 | 5.22 | 3.96 | 6.36 |
| *TB* − 12 ∗ 6 ∗ 6 | 5.5 | 4.60 | 11 | 7.1 | 6.5 | 5.4 | 8.58 |
| *TB* − 12 ∗ 12 ∗ 3 | 3.3 | 3.30 | 7.92 | 5.2 | 5.22 | 3.96 | 6.36 |
| *TB* − 12 ∗ 12 ∗ 6 | 4.34 | 4.60 | 11 | 7.1 | 6.5 | 5.4 | 8.58 |

**Table 4.** The effective pier width using Equation (4) and the calculated scour depth ($Z - CAL$).

| Cases ($W*Lu*T$) cm | $Z-EXP$ (cm) | $a$ (cm), Equation (2) | $Z-CAL$ (cm) [27] | $Z-CAL$ (cm) [43] | $Z-CAL$ (cm) [46] | $Z-CAL$ (cm) [47] | $Z-CAL$ (cm) [33] |
|---|---|---|---|---|---|---|---|
| $RB - 12*6*3$ | 5.1 | 2.98 | 7.14 | 4.68 | 4.88 | 3.57 | 5.76 |
| $RB - 12*6*6$ | 7.8 | 3.95 | 9.48 | 6.16 | 5.87 | 4.74 | 7.52 |
| $RB - 12*12*3$ | 4.1 | 2.98 | 7.14 | 4.68 | 4.88 | 3.57 | 5.76 |
| $RB - 12*12*6$ | 6.52 | 3.95 | 9.48 | 6.16 | 5.87 | 4.74 | 7.52 |
| $TB - 12*6*3$ | 3.8 | 2.53 | 5.64 | 3.72 | 4.19 | 2.82 | 4.50 |
| $TB - 12*6*6$ | 5.5 | 3.05 | 6.48 | 4.26 | 4.58 | 3.24 | 5.22 |
| $TB - 12*12*3$ | 3.3 | 2.53 | 5.64 | 3.72 | 4.19 | 2.82 | 4.50 |
| $TB - 12*12*6$ | 4.34 | 3.05 | 6.48 | 4.26 | 4.58 | 3.24 | 5.22 |

The comparative assessment involved contrasting the maximum experimental scour depth results ($Z - EXP$) against the calculated scour depths derived from various methods [27,33,43,46,47] ($Z - CAL$), as depicted in two scenarios illustrated in Figure 16A–D. Additionally, each equation's results and root mean square error ($RMSE$) relative to the observed value were computed and summarized in Table 5.

Figure 16A–D and Table 5 delineate that the scour equations in [43], particularly with pier equivalent width ($a$) calculated by Equation (4), stand out as the most reliable approaches for measuring scour depth near piers. When compared to equations from other sources, these equations performed better for both $RB$ and $TB$ debris shapes. The estimates were substantially closer to experimental scour depth values, showing improved precision in estimating, as indicated by a reduction in the $RMSE$ values of 0.7 for the $RB$ debris and 0.65 for the $TB$ debris. These equations are more dependable in general and routinely produce superior estimates of scour depth near piers. The rest of the equations consistently generated good estimates closest to the observed values, specifically when applying the equivalent pier width ($a$) (Equation (4)).

**Table 5.** Comparative performance of empirical equations estimating scour depth near piers with different debris types.

| The Empirical Equations | $De$ and $a$ Equations | Performance |
|---|---|---|
| Reference [27] | Equation (3) | Overestimate scour depth for both $RB$ and $TB$ debris, with varying degrees of accuracy ($RMSE$ = 3.6 and 5.3, respectively) |
| | Equation (4) | The trend of overestimation for $RB$ debris ($RMSE$= 2.4), slightly improves $TB$ debris ($RMSE$ = 1.9). |
| Reference [43] | Equation (3) | Alignment with actual values is evident for $RB$ debris ($RMSE$ = 0.85), while there is a notable overestimation for $TB$ debris ($RMSE$ = 1.9). |
| | Equation (4) | $RB$ debris shows close alignment with actual values ($RMSE$ = 0.7), while $TB$ debris displays better estimation precision ($RMSE$ = 0.65). |
| Reference [46] | Equation (3) | $RB$ debris tends to slightly overestimate ($RMSE$ = 0.87), while $TB$ debris shows more significant overestimation ($RMSE$ = 1.6). |
| | Equation (4) | $RB$ debris mildly underestimates ($RMSE$ = 1), whereas $TB$ debris demonstrates improved precision ($RMSE$ = 0.67). |
| Reference [47] | Equation (3) | There is a tendency toward underestimation for $RB$ debris ($RMSE$ = 1.4). Conversely, $TB$ debris shows a more accurate estimation ($RMSE$ = 0.63) |
| | Equation (4) | Tends to underestimate: $RMSE$ = 1.9 for $RB$ debris and $RMSE$ = 1.37 for $TB$ debris. |
| Reference [33] | Equation (3) | Tends to overestimate: $RMSE$ = 1.7 for $RB$ debris and $RMSE$ = 3.2 for $TB$ debris. |
| | Equation (4) | Tends to mildly overestimate: $RMSE$ = 1 for $RB$ debris, and aligns slight overestimation with actual values: $RMSE$ = 0.83 for $TB$ debris. |

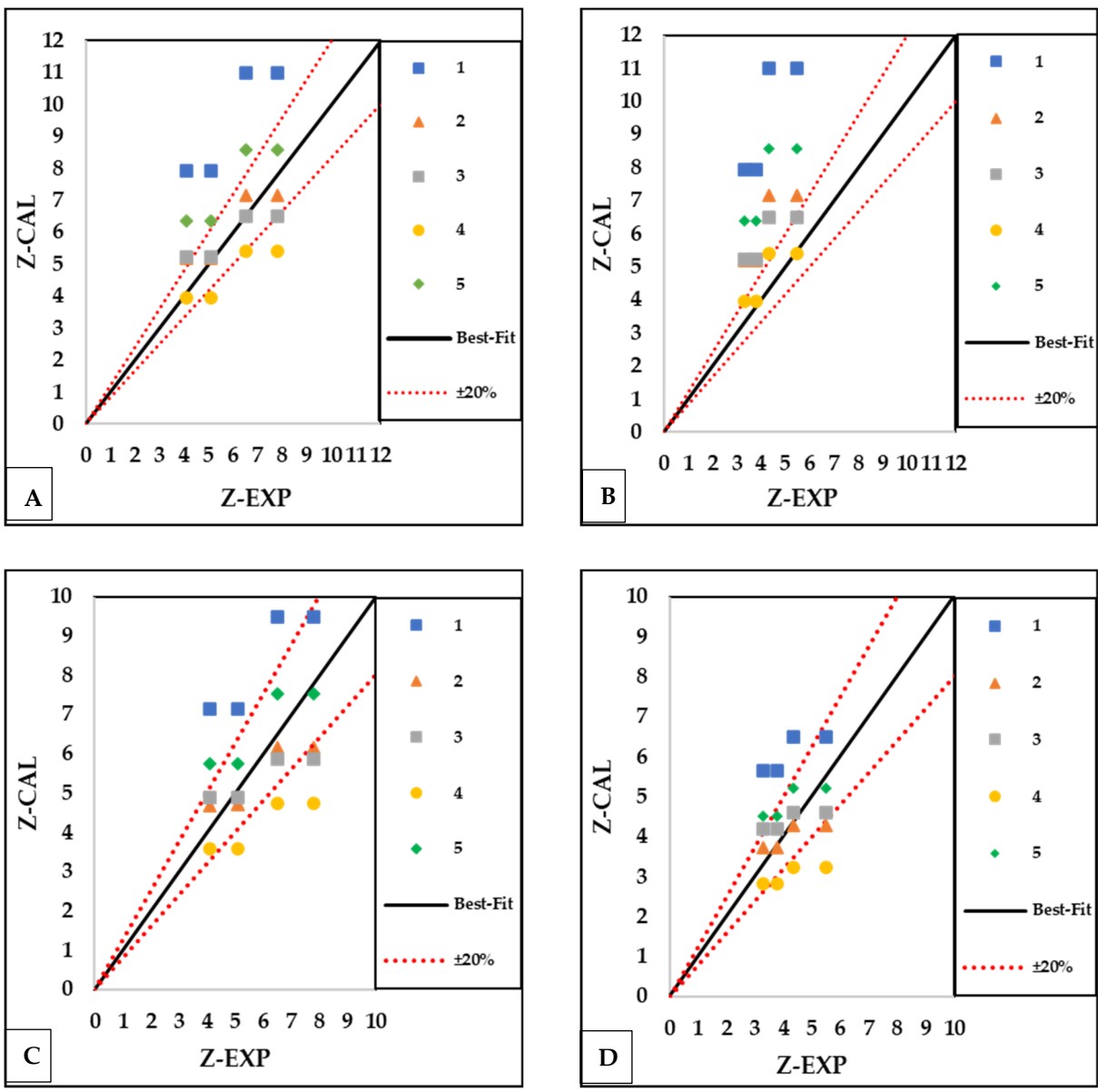

**Figure 16.** Comparison of calculated scour depths ($Z - CAL$) by equations in (1) Melville and Sutherland (1988) [27], (2) Melville and Chiew (1999) [43], (3) the CSU 2001 [46], (4) May, Ackers, and Kirby [47], and (5) Sheppard et al. (2014) [33] after employing the effective pier width (Equation (3)) for (**A**) rectangular debris (*RB*) and (**B**) triangle debris (*TB*), and the equivalent pier width (Equation (4)) for (**C**) rectangular debris (*RB*) and (**D**) triangle debris (*TB*) with measured scour depth ($Z - EXP$).

## 8. Derivation and Application of Debris Correction Factor Model

According to [22], the proposed Equation (5) introduces a novel methodology to analyze the effects of wood debris accumulation on bridge pier scour and provides a design framework for predicting scour depth increments. This equation considers the factors contributing to the augmentation of scour depth due to debris accumulation. Debris accumulation induces flow contraction, a primary driver behind the amplified scour depth. Moreover, the shape and dimensions of the debris, including thickness, width, and streamwise length, significantly influence the scour depth. Furthermore, the transverse cross−sectional geometry of the debris accumulation can impact scour around circular bridge piers.

The equation incorporates three primary parameters: the upstream length of floating debris relative to the pier diameter ($L_u / D$), the downstream length of the debris relative to

its upstream length ($L_d/L_u$), and the obstruction percentage ($A\%$). The equation's effectiveness was systematically evaluated across various debris shapes, including rectangular, triangular, and circular configurations:

$$Kd(cal) = 1 + 0.036 * \mathcal{F}\left(\frac{Lu}{D}\right) * \mathcal{F}\left(\frac{Ld}{Lu}\right) * A\%^{1.5} \tag{5}$$

$$\mathcal{F}\left(\frac{Lu}{D}\right) = 1 - 0.057\left(\frac{L_u}{D} - 3\right) \tag{6}$$

$$\mathcal{F}\left(\frac{Ld}{Lu}\right) = 1 - 0.6\left(\frac{L_d}{L_u}\right) \tag{7}$$

$$Kd(meas) = \frac{Zsd}{Zsn} \tag{8}$$

where $Zsd$ is the scour depth due to the debris effect, and $Zsn$ is the scour depth where no debris is induced.

Figure 17 illustrates the relative measured scour depth of this study across all scenarios of the study, $Kd(means)$, and the prediction results calculated from Equation (5), $Kd(cal)$. Notably, when considering a relative debris thickness ($T/Y$) of 0.25 with an obstruction percentage ($A\%$) of 8.3% (3.47% for the $TY$; low value), there was a notable agreement between the experimental data and the equation's predictions. Similarly, reasonable alignment occurred for scenarios with $T/Y = 0.5$ and $A\% = 16.7\%$ (6.9% for the $TY$; medium value). However, discrepancies arose, particularly in cases involving the $HW$ configurations with downstream extensions, attributed to the lack of inclusion of the high wedge debris shape in a prior investigation [22]. This discrepancy persisted for most debris scenarios when analyzing the scenarios with $T/Y = 1$ and $A\% = 33.3\%$ (16.7% for the $TY$; high value), where the debris rested on the bed. In these instances, applying the equation to conditions with $A\%$ exceeding 16% led to errors exceeding 20%, highlighting its limitation for scenarios with higher obstruction percentages.

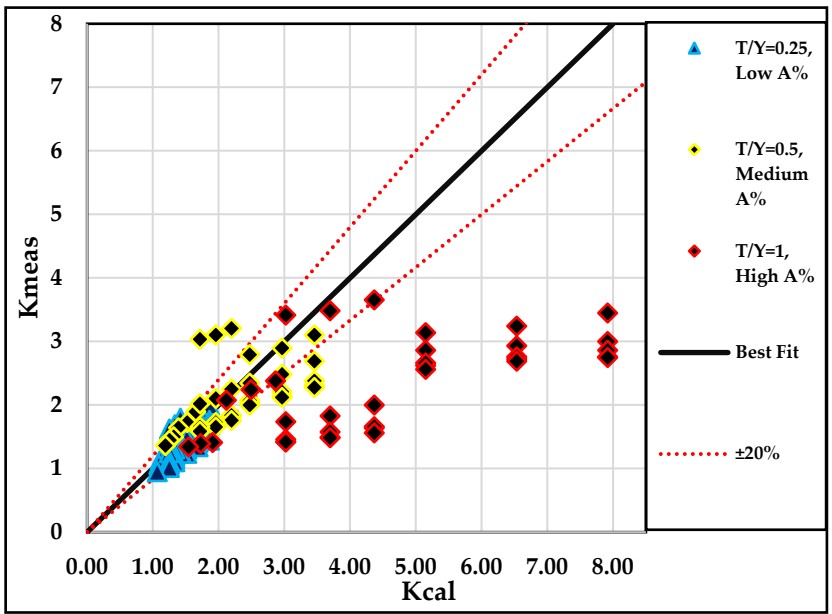

**Figure 17.** Comparative analysis of measured scour depths ($Kd(means)$) and predicted results ($Kd(cal)$) using Equation (5) across various scenarios in this study.

To address limitations in the previously established equation for scenarios exceeding a blocking ratio ($A\%$) of 16% and to account for the impact of different debris shapes, this research introduced a novel debris shape factor ($S.F_d$). This factor incorporates the

relative scouring potential of various debris shapes compared to a reference shape (typically rectangular block−shaped debris). The $S.F_d$ is defined by Equation (9):

$$S.F_d = \frac{Zsd_s}{Zsd_B} \tag{9}$$

where $Zsd_s$ is the scour depth due to the specific debris shape under consideration, and $Zsd_B$ is the scour depth due to the block−shaped debris (rectangular debris).

Equation (10) builds upon this concept by basing its formulation on the experimental findings. This equation was applied to the entire experimental dataset, encompassing obstruction ratios above 14%, and was supplemented with data from previous studies [23] to validate its effectiveness. Equation (10) yielded the best predictions for all the conditions and shapes of debris studied here, even when the effect of the debris moving downstream from the pier was considered, as depicted in Figure 15. The equation is structured as follows:

$$Kcal = a + A\%^b \times S.F_d{}^c \times \mathcal{F}\left(\frac{Ld}{Lu}\right)^d \times \mathcal{F}\left(\frac{Lu}{D}\right)^e \tag{10}$$

where $a, b, c, d,$ and $e$ represent coefficients determined through multiple regression analysis, yielding values of 0.31, 0.54, 1.26, 0.33, and 0.97, respectively.

Equation (10) incorporates the debris shape factor ($S.F_d$), obstruction ratio ($A\%$), and geometric parameters, such as the downstream extension of debris relative to its upstream length ($\mathcal{F}\left(\frac{Ld}{Lu}\right)$) Equation (7), and the upstream length of floating debris relative to the pier diameter ($\mathcal{F}\left(\frac{Lu}{D}\right)$) Equation (6). These coefficients were derived from statistical analysis to optimize the equation's predictive accuracy. The agreement between Figure 18 and the experimental dataset was deemed excellent, supported by a coefficient of determination ($R^2$) of 0.86 and a $RMSE$ of 0.2, indicating strong correlation and minimal prediction error.

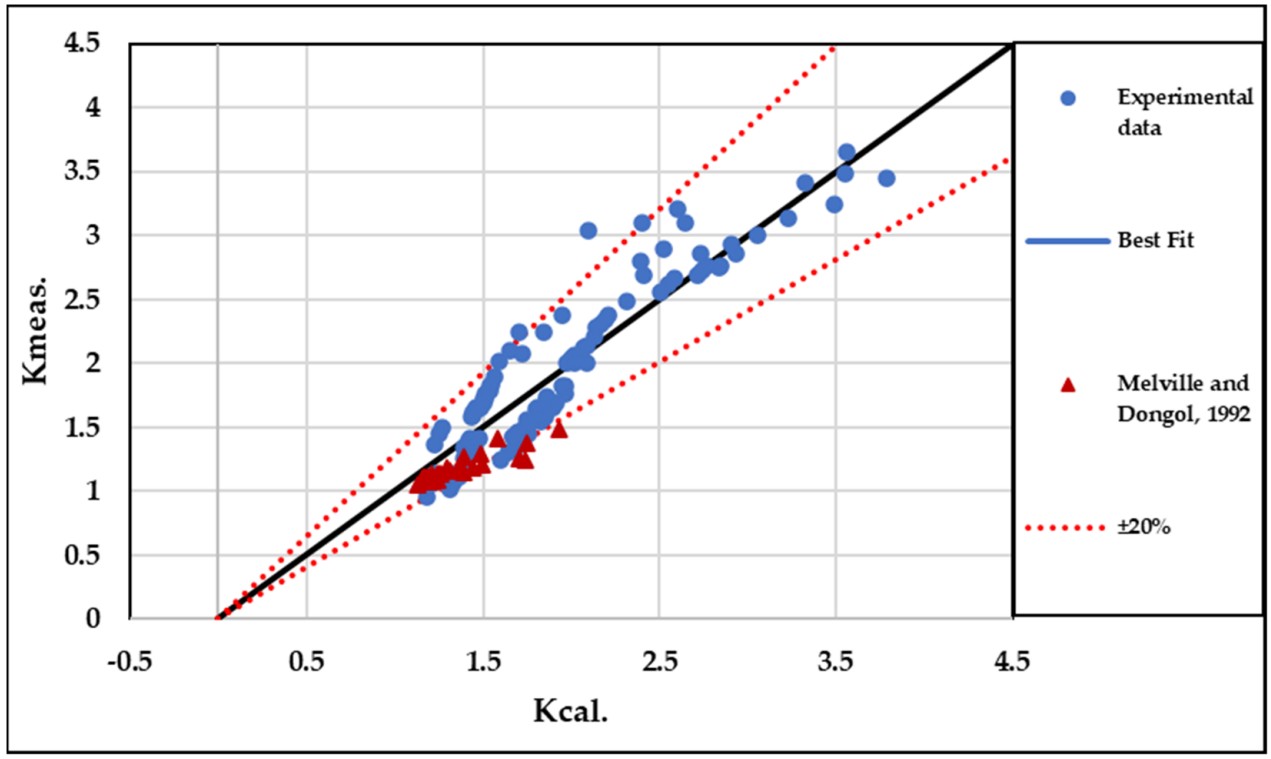

**Figure 18.** Comparative analysis of measured scour depths ($Kd(means)$) and predicted results ($Kd(cal)$) utilizing Equation (10) across various scenarios in the present study and the study by Melville and Dongol [23].

## 9. Conclusions

This study emphasized the influence of debris shape and its downstream extension on the maximum scour depth around a single cylindrical pier. It incorporated six distinct types of the *LWFD*. The experiments were carried out in a rectangular channel with a uniform flow of clear water and a bed composed of homogeneous sand with a particle diameter ($D_{50}$) of 0.93 mm. The key findings are summarized below:

1. The shape of the debris mass influenced the scour depth more than its size. High wedge (*HW*) debris near the pier produced the deepest scour, while triangle yield (*TY*) debris resulted in a minimal scour depth. Understanding these dynamics is crucial for effective mitigation strategies around bridge piers.

2. Deeper scour holes and higher deposition heights were observed when debris rested entirely on the streambed (highest blockage percentage, *A*%).

3. The ratio of the *LWFD* length to its width ($L_u/W$) significantly influenced the shape of scour holes in both longitudinal and transverse profiles, especially when the $L_u/W = 0.5$.

4. Downstream extension of debris ($L_d/L_u$) can reduce the scour depth directly below the pier, possibly due to flow deflection. However, it can also lead to a larger overall scour hole area.

5. The study proposed a novel debris correction factor model (Equation (10)) that considered the debris shape (*S.Fd*), obstruction ratio (*A*%), and other geometric parameters. This model demonstrated good agreement with experimental data and can be used to improve scour depth predictions in scenarios with debris accumulation.

6. Debris presence significantly altered the dimensions of the scour hole, with rectangular debris causing a substantial increase in area and volume. The extension of debris downstream primarily increased the scour area rather than affecting the scour depth across different debris shapes.

## 10. Limitations and Further Research

The current study acknowledges several limitations that warrant consideration when interpreting the findings and their practical applications:

1. The study's reliance on a narrow flume width may limit its representation of larger piers and debris behavior, potentially leading to inaccuracies in assessing scouring effects during severe floods.

2. Although assigning a uniform debris width aimed at optimizing lateral scour, the study acknowledges potential impacts on scour hole dimensions, particularly when the effective width exceeds 10% of the flume (*B*).

3. When more than one−third of the flow breadth is restricted, local flow velocity effects may differ, potentially influencing scouring trends around the pier.

4. Due to the formidable challenge of developing a suitable debris model with appropriate density, the study did not fully explore the dynamic interaction between debris and flow.

5. Maintaining a constant width (*W*) for the debris triangle yield (*TY*) shape ensures consistency but may lead to variations in wall inclination, potentially affecting flow dynamics.

6. The study established an empirical equation developed from limited experimental datasets. Therefore, it may not adequately capture the complexity of real−world scenarios, necessitating careful analysis of the findings.

Due to the insufficient available data and considering the significance and economic implications associated with bridge failures, additional research efforts should be conducted on debris accumulation around bridge piers.

**Author Contributions:** Conceptualization, M.A.-J. and R.P.R.; methodology, M.A.-J. and R.P.R.; formal analysis, M.A.-J.; investigation, M.A.-J.; data curation, M.A.-J.; writing—original draft preparation, M.A.-J.; writing—review and editing, M.A.-J. and R.P.R.; supervision, R.P.R.; project administration, R.P.R. All authors have read and agreed to the published version of the manuscript.

**Funding:** This research was funded by Széchenyi István University.

**Data Availability Statement:** All of the data are available in the paper.

**Conflicts of Interest:** The authors declare no conflicts of interest.

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
