# Peer review of "Hydrodynamic Modeling and Comprehensive Assessment of Pier Scour Depth and Rate Induced by Wood Debris Accumulation"

_hydrology, doi:10.3390/hydrology11040052_

Round 1
Reviewer 1 Report
Comments and Suggestions for Authors
This paper addressed an interesting subject in pier-scour by flow accompanied with floating debris. A clear aim and methodology of the study were introduced. The experimental setup was acceptable except the large ratio of the debris width and the flume width. The scour depth and scoured morphology were measured with care under the impacts of many different debris formations. As a final result, a new equation estimating the effective width of pier scour was formulated, which is probably practical in the specific events similar to that modeled in the experiments.
In addition to the debris width and length paid a focus on in the paper, many other factors are combined to influence the pier scour, such as those of hydrodynamics, Froude Number and so on, those of sediment size, etc. I appreciate the authors’ serious-minded consideration for the limitations of this paper. The new (modified) equation was simply presented in polynomials by fitting the results calculated by the equation from the literature [35] and the measured in this study. The paper may be highly worth reading if it be fundamentally improved upon the contributions from the literature, for example, challenging the limitation 2, or 3 explained in the end of the paper.
In the conclusions, the one with label 7 is a common sense for the readers who are concentrated on the pier scour; the one with label 13 is irrelevant because the effects of the debris downstream extension on the pier lateral loading is not found in the context.
Author Response
Dear Reviewer,
The authors would like to first thank the reviewer for their time, effort, and suggestions to improve the manuscript.
As general information, there have been substantial additions to the text to better explain points of discussion, results, etc. They are highlighted in yellow in the corrected manuscript. There are also editorial revisions on grammar and structure in several places, which may not be highlighted in yellow but were intended for clarification of a sentence or phrase. The second author (a native English speaker/writer) has thoroughly re-examined the entire text.
- We acknowledge your concern regarding the large ratio of debris width to flume width in our experimental setup. This is a limitation, and we will strive to address it in future studies by using a wider flume or employing smaller debris elements that maintain a more realistic scale. However, the debris dimensions were chosen carefully, considering real-life debris sizes. The width of debris, W, was expected to affect mainly the lateral (transverse) extent of scour and have minimal influence on the maximum scour depth, according to [1]. Furthermore, we have found many research references with similar dimensions. Table 1 lists four other studies with similar B/W ratios and others with only slightly better ratios. Certainly, the size ratio for our study was within the range.
- The updated proposal includes a new debris correction factor model, shown as Equation 10. This model takes into account things like the shape of the debris (S.Fd), the wide range of obstruction ratios (A%), and different geometric parameters. The validity of this model has been checked against experimental data, showing strong agreement. It may be better at predicting scour depth in situations where debris builds up.
- Finally, we thank you for your feedback on the conclusions section of our paper. We agree that clarity and relevance are essential, and we will ensure that any future iterations of the manuscript address these concerns appropriately. We agree that (7) is common sense for specialists, but perhaps non-specialists would read it as well. We can strike it from the list. (13) is removed.
Once again, we appreciate your thorough review and valuable feedback. Your insights will undoubtedly contribute to the enhancement of our research.
Kind regards
Dr. Richard P.Ray, Muhanad Al-Jubouri
[1] P. F. Lagasse, Effects of debris on bridge pier scour, vol. 653, Transportation Research Board, 2010.

Reviewer 2 Report
Comments and Suggestions for Authors
General comment:
The Authors analyzed an interesting phenomenon conducting novel experiments aiming at investigating some effects on scour processes caused by the presence of a large debris accumulation. I read the paper with interest. However, in the present form, it is very far from being ready for publication. Namely, the physical mechanisms should be better described providing more theoretical and physical insights on the analyzed phenomenon. Furthermore, the elaboration of experimental data is not clear. Considering the novelty of experimental tests, at this stage, I recommend major/major revisions. Nonetheless, I think that a very big effort is needed to improve the paper. Several portions need to be re-structured/re-written.
Specific comments:
1) Lines 63-64: it is not clear to me what Authors mean by “The investigation further…circumstances”. Please clarify.
2) Please check the journal format and verify if symbols should be in italic in the text. In addition, please pay attention to subscripts and capitol letters (e.g., K in eq.2 and k in line 98).
3) Overall, the introduction is too long and should be reduced by keeping essential information.
4) Figure 2: this figure is useless since its content is already present in the text. Please remove it.
5) Following my comment 3, in section 2, lines 174-234 report well-known results. Consequently, this portion can be significantly shortened.
6) Lines 237-242: these statements were repeated several times. Please avoid repetitions. (Please note that this is just an example.)
7) Lines 265-275: please indicate the dimensions of different debris. This can be done also in the Figure 3.
8) Lines 331-324: it is not clear if continuous measurements were also taken with laser scanner. If so, can you indicate the type of laser scanner that was employed?
9) Line 351: I believe that Authors want to mean frontal area here. Please clarify.
10) As for the debris TY, by keeping W constant, the inclination of the side wall of the debris should change. Consequently, are such configurations comparable considering that the flow dynamics may be affected by this parameter? Please clarify.
11) Figure 7: the caption of the figure does not reflect the presented image that refers to the scour configuration at equilibrium. Please check and revise. Furthermore, please clarify the units for the scale on the right side of the figure.
12) Figure 8: apparently, there is an inversion of the general trend for HW and T=12_W=12_Lu=12. Can authors provide clearer justifications corroborating such experimental evidence?
13) Lines 413-429: these explanations are very qualitative and not clear.
14) Lines 432-458: authors limited themselves to describe the content of the figures without providing real insights on the physics of the phenomenon. Furthermore, I have doubts on the proposed comparison. Considering that one of the main parameters affecting the phenomenon is the blockage percentage, I think that such comparisons are significantly affected by the specific values of that parameter. The presented results do not allow to highlight such effect. I suggest showing the effect of A% instead.
15) Line 465: this statement corroborates what I mentioned in my previous comment. Overall, it seems to me that few real insights are provided. In other words, the analysis of the results appears to be very qualitative and descriptive. Finally, it is also not clear in several rooms.
16) Lines 466-476: I do not understand how these lines are related to the statement at line 465. There are no general trends here.
17) Lines 477-479: this trend is due to the fact that A% increases with T, but it is not visible from any graph.
18) Figures 12, 13 and 14: please see my previous comments.
19) Lines 547-549: Apparently, authors state that none of existing methodology explicitly considered the effect of debris accumulation on scour depth. If I correctly understood this sentence, I do not agree with it.
20) Line 542: what do you mean by observed effective width. Could you please clarify how you observed this parameter?
21) Figure 15: it is not clear which experimental data are contrasted against predicted ones.
22) Lines 578-582: it is not at all clear why authors selected those specific ranges of variation of parameters, considering that they tested larger ranges according to Table 1. Consequently, I do not understand what these equations really represent. Please explain.
23) Figure 16: this figure is conceptually not correct since it is hybrid. Namely, the parameter T is dimensional whereas Kd meas and Kd cal are non-dimensional. I wonder how authors can generalize their results.
24) Lines 605-618: It is not clear what authors want to state here. In particular, I do not understand what Figure 18 really represents. Moreover, what do you mean by “Where a, b, c, and d are coefficients…from empirical Fig. (20)”? Please clarify.
25) Again, section 8 is not clear.
26) Overall, the elaboration of results needs to be substantially improved. In the present form, it is not at all clear. I invite authors to carefully re-structure this part.
27) Finally, the English of the manuscript should be substantially improved. Similar comments for some figures.
Comments on the Quality of English LanguageThe English of the manuscript should be substantially improved
Author Response
Dear Reviewer,
The authors would like to first thank the reviewers for their time, effort, and suggestions to improve the manuscript.
As general information, there have been substantial additions to the text to better explain points of discussion, results, etc. They are highlighted in yellow in the corrected manuscript. There are also editorial revisions on grammar and structure in several places, which may not be highlighted in yellow but were intended for clarification of a sentence or phrase. The second author (a native English speaker/writer) has thoroughly re-examined the entire text.
- Lines 63–64: We apologise for the unclear statement. This text has been replaced.
- We worked specifically on symbols and made them appropriately formatted according to the journal's guidelines.
- We have conflicting suggestions from reviewers, where one asks for more and another asks for less. We have done our best to condense and streamline the introductory remarks.
- We remove figure 2 as its content duplicates information already presented in the text.
- We were again trying to reconcile conflicting reviews. Many of these sections were rewritten for better clarity and made shorter.
- We eliminated repetitions and ensured conciseness in the manuscript.
- While we re-drawed the figure to include the debris dimensions, we also included the dimensions of different debris, adding this information to the text before figure 2 for clarity.
- We clarified that continuous measurements were taken with a laser scanner, and we added a better description of the measurement system.
- We clarified the intended meaning, particularly regarding the terms submerged frontal area (Ad) and total fontal area (Ao).
- We acknowledge the concern raised regarding the potential impact of maintaining a constant width (W) for the debris TY shape on the inclination of the side wall and its potential effect on flow dynamics. We opted for this approach to ensure consistency across different pier shapes and to maintain a fixed obstruction ratio (A%) by maintaining fixed dimensions (W, Lu, and T) for all debris shapes. We chose to prioritise continuity in our study. That may be said for all of the shapes to some degree. We had a finite number of shapes and tests and chose to configure them in this manner. Any shape is “different” when it changes any dimension. We chose to “anchor” the near surface dimensions and grow the shapes from there. Our reasoning was that the debris clump could be thought to grow downward in such a way. We appreciate your highlighting this limitation, and we will duly acknowledge it in the limitation section of our manuscript for future studies.
See, for instance, ref. [1].
- We revised the figure 7 caption to accurately reflect the presented image and clarify the units for the scale.
- The sudden change in direction shown in Figure 8 is due to the wedge debris surface becoming more inclined perpendicular to the flow direction. This is especially clear for HW with T = 12, W = 12, and Lu = 12. This inclination angle tends to rise as the debris length increases from 6cm to 12cm, thereby inducing alterations in flow dynamics. In comparison to HW with (W = 12, Lu = 6, and T = 12), this change in flow dynamics results in a discernible but slight elevation in scour depth. The intensified plunging flow towards the pier is the primary contributor to this effect. Notably, this outcome aligns with findings in other experimental studies [1]–[3] and the above reference.
- We have added discussion to this part of the paper. We are not sure how explanations for the causes of behaviour could be anything but qualitative since we did not measure other flow or pressure forces.
- We strived to provide deeper insights into the physics of the phenomenon and consider the effect of blockage percentage on comparisons. More attention has been given to the A% effects in the figures. There is also discussion on the effects of A% in the following sections on the derivation and application of the debris correction factor model. Eq. (9), and (10) and Figures 18.
- We have endeavoured to expand our discussion of results in the revised version. This was an experimental study, and we are continuing our work on analysis and modelling.
- We revised the analysis of the results to provide more meaningful insights and improve overall clarity.
- We ensured that these lines are appropriately related to the preceding statement and provide clear explanations of any observed trends. We clarified the relationship between the observed trend and the increase in obstruction ratio (A%).
- We addressed issues with Figures 12, 13, and 14 based on your previous comments.
- We meant that all of the above equations from 1 to 5, which were originally formulated to calculate the scouring depth in cases where the effect of debris is not present,. However, we revised this statement and ensured accuracy regarding existing methodologies. The statements were unclear, and this passage has been rewritten.
- Sorry, it was a typo error. We clarified the concept of effective and equitant width and provided a clear explanation of how this parameter was determined.
- We are aligning the test data and prediction equations better. The initial comparisons used our experimental data that best fit the assumptions of the predictive equations, hence the reduced number of experimental values. Later, with the proposed equation modifications, we used the entire experimental data set. We clearly indicated which experimental data are contrasted against predicted ones in tables 4 and 5.
- We apologise for any misunderstanding. Our intention was to explain the specific ranges of parameter variation outlined by the original researcher, who formulated the equations and provided an interpretation. To enhance clarity, we have refined the presentation of these parameters within the following rows: 572–582.
- We ensured conceptual correctness and clarified the dimensional and non-dimensional aspects of the figure. However, this graph is not intended for generalization; rather, it is intended to show that there is a depth (T) effect that the equation does not account for. A discussion has been added to clarify that, and the figure was modified.
- We provided different, clearer explanations and ensured understanding coefficients (see rows 606–629).
- We worked on improving the clarity of this section.
- We carefully restructured the elaboration of results to improve clarity and coherence.
- We thoroughly reviewed and improved the manuscript's English language quality, as well as addressed any issues with figures.
Thank you for your valuable feedback, and we are committed to addressing these points to enhance the quality and readiness of our manuscript for publication.
References:
[1] I. Schalko, C. Lageder, L. Schmocker, V. Weitbrecht, and R. M. Boes, “Laboratory flume experiments on the formation of spanwise large wood accumulations: Part II—Effect on local scour,” Water Resour. Res., vol. 55, no. 6, pp. 4871–4885, 2019.
[2] H. Hamidifar, D. Mohammad Ali Nezhadian, and I. Carnacina, “Experimental study of debris-induced scour around a slotted bridge pier,” Acta Geophys., vol. 70, no. 5, pp. 2325–2339, 2022.
[3] P. F. Lagasse, Effects of debris on bridge pier scour, vol. 653, Transportation Research Board, 2010.
Kind regards
Dr. Richard P. Ray, Muhanad Al-Jubouri

Reviewer 3 Report
Comments and Suggestions for Authors
Review comment (2024-02-22)
Title: Hydrodynamic Modeling and Comprehensive Assessment of Pier Scour Depth and Rate Induced by Wood Debris Accumulation
This paper addresses a significant concern in civil infrastructure resilience by investigating the combined effects of various debris shapes on predictive scour depth models around non-cylindrical bridge piers. The research fills a gap in prior literature by delving into a less-explored area, shedding light on the complex dynamics governing local scour phenomena. The experimental approach adopted in this study is commendable, allowing for a systematic exploration of factors such as pier geometries, debris arrangements, and submersion depths. The inclusion of various pier shapes— cylindrical, square, etc.—provides comprehensive insights into their respective influences on scour patterns.
1) Major revision
- I agree that your manuscript is different from your previous paper in the Sustainability- MDPI, even you used the same experimental setup and apparatus. Of course, although your manuscript has significant differences from the content of "https://doi.org/10.3390/su152215910", I doubt whether the content is so different that it does not need to be included as a reference. Please respond once again.
- I also recommend you add some of key results and conclusions in the abstract part of the manuscript.
- I am suggesting the additional reference “DOI: 10.1061/(ASCE)HY.1943-7900.0000289”. And you can compare this paper and [34]. You can also check the reports and paper from the Deltares in Nethlands such as Scour manual by Hoffmans and Verheij (https://doi.org/10.1201/b22624).
- Of course, there are several limits to experimentally implementing the theoretical equilibrium state of bridge scour. However, it is regrettable that more extended experimentation periods were not available recently compared to the durations of the experimental studies conducted in [36] or [37], which could have provided clearer insights into the dimensions of the scour holes reaching equilibrium states. I suggest you the future study will be conducted for longer periods with automatic measuring system.
2) Minor revision
- Figures and tables should be modified
- In line 39: Using the acronym "LFWD" to abbreviate "big floating wood debris" may not be appropriate as it could obscure understanding and clarity. Instead, it's recommended to use the full term "big floating wood debris" or consider alternatives such as "floating wood debris" or "large floating debris" depending on the context, as they are clearer and more easily understood. Surely you used the Large Wood Floating Debris in line 237. Therefore, you can modify the line 39. Also, you mentioned LWFD in the chapter 9 as “Large Wood and Debris” in line 650.
- In all of the equations, parameters should be same in the definition including subscript or superscript and italic fonts.
- Please confirm whether I should consistently use a space between numbers and units or keep them directly adjacent.
- Please add the length units in Figure 7 and 11.
- Please rearrange the legends and axis-titles in Figure 8, 10, 13, 14, 15, 17, 20.
- Why does it start from number 6 in line 653?
- Slope of the scour hole can me expressed with “steep and mild” in line 453.
- Expression or notations of the all the parameters used in the equations should be same with those in the text.
Addressing these aspects would enhance the academic value and practical applicability of this paper.
Comments on the Quality of English LanguageThis manuscript demonstrates a high level of proficiency in English composition. The language used is sophisticated and precise, with complex sentence structures and technical vocabulary appropriate for an academic research context. The writer effectively communicates intricate concepts related to debris-induced local scour around cylindrical bridge piers, including the methodology employed, specific variables investigated, and key findings. Overall, the writing reflects an advanced level of English writing proficiency indicative of someone experienced in academic discourse and scientific research. However, it is disappointing to note that your research content or citations do not reflect the same level of proficiency as your writing. Furthermore, submitting research content that has been fragmented or duplicated inappropriately is highly unethical. I am very sorry to tell you this.
Author Response
Dear Reviewer,
The authors would like to first thank the reviewers for their time, effort, and suggestions to improve the manuscript.
As general information, there have been substantial additions to the text to better explain points of discussion, results, etc. They are highlighted in yellow in the corrected manuscript. There are also editorial revisions on grammar and structure in several places, which may not be highlighted in yellow but were intended for clarification of a sentence or phrase. The second author (a native English speaker/writer) has thoroughly re-examined the entire text.
Major revision
- We acknowledge the reviewer's concern regarding the similarity between our manuscript and the previously published work in Sustainability-MDPI. While both studies utilise only a similar experimental setup and apparatus, it's crucial to highlight the substantial differences in the research focus, methodology, and findings between the two papers. This study delves into the intricate influence of lengthwise floating debris accumulation, extending downstream from the circular pier, covering up to 66% of the debris length downstream of the pier, a less-explored area in the literature. The primary focus is on assessing its impact on the final scour depth, hole volume, and area. As part of this study, a new empirical equation is created using important factors like the depth of submergence debris, the percentage of debris that blocks the flow (A%), debris shape, and the ratio of the length of upstream to downstream debris extension. While our previous work, which primarily explored the effects of floating debris situated just upstream of different pier shapes on scour hole depth,. In that earlier study, we used different sediment particle sizes, flow conditions, and pier sizes in order to show how pier shape factors and debris characteristics affect each other and how that affects the final scour depth value. We revised the manuscript to clearly articulate these distinctions and emphasise the novelty of our work compared to the previous publication.
- We appreciate the suggestion to include key results and conclusions in the abstract section. We revised the abstract accordingly to provide a concise overview of the significant findings of our study.
- We incorporated the suggested references (by Pagliara) and compared them. Additionally, we explored relevant reports and papers to enhance the theoretical background.
- We agree that longer measuring times are better, but we have found numerous references with similar testing times (4–8 hours). We did run some longer tests (72 hours.) that did not produce any further insight, and the time spent had only a barely measurable change.
Minor revision
- We modified figures and tables as per the reviewer's suggestions to improve clarity and consistency.
- We used the full term for large floating wood debris (LFWD). We also enhanced the consistency throughout the manuscript.
- We corrected parameters in equations, including subscript, superscript, and italic fonts, for clarity and consistency.
- We will review the spacing between numbers and units throughout the manuscript and make necessary adjustments for consistency.
- Length units were added to the figures.
- Legends and axis titles were rearranged for better readability and clarity.
- The numbering in the conclusion was corrected.
- We expressed the slope of the scour hole as "steep and mild" for clarity and consistency with terminology used elsewhere in the manuscript.
- Parameters used in equations were modified to be consistent with those in the text
Comments on the Quality of the English Language:
We honestly don’t know what to make of this statement. The second author is a native English writer and speaker and has 52 publications listed in Scopus with 470 citations. He has also reviewed countless journal, conference, and textbook publications. One would expect the level of discourse in this paper to be above average.
The research content is experimental and performed by the first author under the second author's supervision (as well as other collaborating professors). This is not fragmented (whatever that means) or duplicated. It is original research performed over many months in the laboratory.
One can always sit back and poke holes at experimental research; “you should have done X instead of Y”, “why didn’t you apply my favorite equations to the analysis instead of those other authors” etc. But that is hardly the point. This paper is presenting the results of our study, nothing more, nothing less.
Kind regards
Dr. Richard P.Ray, Muhanad Al-Jubouri

Round 2
Reviewer 2 Report
Comments and Suggestions for Authors
I revised the paper and found that it has been improved. My main concerns were adequately addressed by authors.
However, there are still some figures that require improvements (e.g., see the legend of Fig. 16). Likewise, please check again all symbols (in particular subscripts!)
Overall, I believe that the paper can be accepted after these minor revisions.
Author Response
Thank you for your constructive feedback. We have carefully reviewed Fig. 16 and its legend, and we have made necessary improvements to ensure clarity and accuracy. Additionally, we checked all symbols, with particular attention to subscripts, to ensure consistency and correctness throughout the paper.
If you have other comments, please contact us.
We appreciate your suggestions.
kind regards
Reviewer 3 Report
Comments and Suggestions for Authors
I truly appreciate your efforts. My previous review opinion is that I hope that your manuscript will have a valuable impact on many researchers in the future. I sincerely hope you don't misunderstand.
I also ask you to rearrange and revise the legends of each Figure (They overlap so we can't see them, or their positions are slightly off).
- Figure 8, 10, 12, 15, 16, 17, 18
Author Response
Thank you for your kind words and encouragement.
We value your feedback and understand your concerns regarding the legends of Figures 8, 10, 12, 15, 16, 17, and 18. We revised and rearranged the legends to ensure they do not overlap and are properly positioned for clarity. Your attention to detail is appreciated, and we are committed to ensuring the quality of our manuscript.
Kind Regards